# Rethinking Exploration in Reinforcement Learning with Effective Metric-Based Exploration Bonus

**Yiming Wang[1], Kaiyan Zhao[1,2], Furui Liu[3], Leong Hou U[1]***

[1]State Key Laboratory of Internet of Things for Smart City, University of Macau, Macao SAR, China
[2]School of Computer Science, Wuhan University, Wuhan, China
[3]Zhejiang Lab, Hangzhou, China
`wang.yiming@connect.um.edu.mo,zhao.kaiyan@whu.edu.cn`
`liufurui@zhejianglab.com,ryanlhu@um.edu.mo`

## Abstract

Enhancing exploration in reinforcement learning (RL) through the incorporation of intrinsic rewards, specifically by leveraging *state discrepancy* measures within various metric spaces as exploration bonuses, has emerged as a prevalent strategy to encourage agents to visit novel states. The critical factor lies in how to quantify the difference between adjacent states as *novelty* for promoting effective exploration. Nonetheless, existing methods that evaluate state discrepancy in the latent space under $L_1$ or $L_2$ norm often depend on count-based episodic terms as scaling factors for exploration bonuses, significantly limiting their scalability. Additionally, methods that utilize the bisimulation metric for evaluating state discrepancies face a theory-practice gap due to improper approximations in metric learning, particularly struggling with *hard exploration* tasks. To overcome these challenges, we introduce the **E**ffective **M**etric-based **E**xploration-bonus (EME). EME critically examines and addresses the inherent limitations and approximation inaccuracies of current metric-based state discrepancy methods for exploration, proposing a robust metric for state discrepancy evaluation backed by comprehensive theoretical analysis. Furthermore, we propose the diversity-enhanced scaling factor integrated into the exploration bonus to be dynamically adjusted by the variance of prediction from an ensemble of reward models, thereby enhancing exploration effectiveness in particularly challenging scenarios. Extensive experiments are conducted on hard exploration tasks within Atari games, Minigrid, Robosuite, and Habitat, which illustrate our method's scalability to various scenarios. The project website can be found at https://sites.google.com/view/effective-metric-exploration.

## 1 Introduction

Reinforcement learning (RL) has made significant strides, yielding breakthroughs across various domains, including video gaming [43], autonomous driving [65], and robotic control [1, 37]. However, for many real-world tasks, defining a dense reward function is non-trivial, yet a sparse reward function based on success or failure is directly available, which makes learning effective policies difficult, as they demand efficient exploration of the state space, highlighting exploration in sparse-reward environments as a core challenge in RL [59].

Methods that quantify the state discrepancy between adjacent steps using specific measures within different metric spaces have shown remarkable success in tasks characterized by sparse rewards. These methods, by leveraging a measure of state discrepancy as an exploration bonus, facilitate agents

---

*Corresponding author.

38th Conference on Neural Information Processing Systems (NeurIPS 2024).

in discovering novel states. For instance, RIDE [51] employs the difference between two consecutive state embeddings, measured under $L_2$ norm, as an exploration bonus. Similarly, NovelD [69] introduces a bonus based on the state discrepancy, as represented by the RND [11] bonus, but under the $L_1$ norm. However, the effectiveness of these $L_p$ norms-based methods is contingent on the scaling factor expressed as the episodic count term $N_{ep}(s)$, which is the number of times state $s$ has been visited during the current episode. The episodic term becomes ineffective when each state is unique and cannot be counted, posting a significant limitation when it comes to more complex, dynamic, and noisy environments [32]. LIBERTY [63] aims to surmount this hurdle by evaluating state discrepancy through the bisimulation metric [25], which links state differences to value differences within the bisimulation metric space, thus incentivizing exploration of novel states with greater value differences and substantially enhancing learning efficiency. Nevertheless, due to the high computational cost or infeasibility of accurately computing bisimulation metrics, the approximation, and relaxations over the metric [13, 68, 63] have been used to optimize efficiency. Our analysis reveals that approximation gap in LIBERTY might break the theoretical guarantees of the bisimulation metric, potentially undermining exploration performance. Furthermore, the efficacy of bonuses based on state discrepancies declines in *hard exploration* tasks or scenarios where state differences are minimal, such as in the "Noisy-TV" [11] problem and vision-based real indoor environments [60], posing a significant constraint on the scalability of these methods.

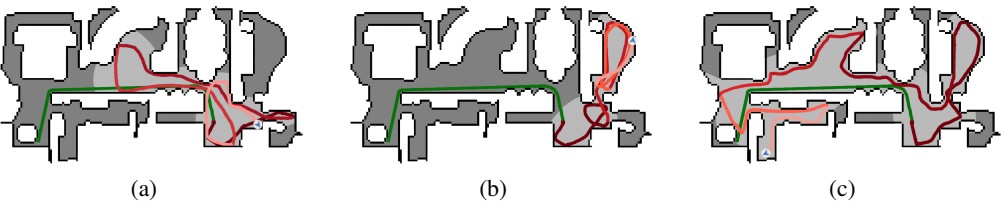

|         (a)         |         (b)         |         (c)         |

Figure 1: Trajectories of policies trained with different exploration algorithms in the real-life indoor environment. (a) episodic count-based method under $L_p$ norm (b) bisimulation metric-based method (c) our method.

To address this aforementioned limitations, we introduce the Effective Metric-based Exploration-bonus (EME) for exploration. Our method features a more resilient metric for evaluating state differences, simultaneously eliminating approximation gap with a theoretical guarantee. EME significantly boosts learning efficiency by forging a closer connection to the value differences between states, while maintaining fidelity to its theoretical framework. Furthermore, we have developed a diversity-enhanced scaling factor to augment exploration efficacy in *hard exploration* tasks, where state differences are notably subtle. Our scaling factor dynamically adjusts the exploration bonus based on the variance of predictions from an ensemble of reward models, which is higher when agents encounter novel state space, thereby encouraging more effective exploration. As illustrated in Figure 1, we present the trajectories of policies trained using count-based state discrepancy bonus, bisimulation-based exploration bonus, and our method within a real-life indoor environment [15] (Full results in Figure 9 of Appendix C.3). Our method successfully explores a significantly larger portion of the space compared to the others, demonstrating the superior effectiveness of EME.

The main contributions of this paper are as follows. Firstly, we conduct a comprehensive analysis of the limitations inherent in current metric-based state discrepancy methods for exploration. Based on the analysis, we introduce an effective metric for evaluating the behavioral similarity between states supported by theoretical assurances. Secondly, we propose a diversity-enhanced scaling factor for exploration bonuses based on the variance of predictions from an ensemble of reward models, which is scalable and effective in *hard exploration* tasks. Lastly, extensive experiments are conducted in Robosuite, Atari games, MiniGrid, and Habitat. The results demonstrate that our algorithm can effectively enhance exploration and accelerate training across various environments.

## 2   Background

We focus on exploration bonuses to incentivize exploration in reinforcement learning (RL). To foster the reader's understanding, we first introduce standard notation and common practices.

**Markov Decision Processes.** We assume the underlying environment is a Markov decision process (MDP), defined by the tuple $\mathcal{M} = (\mathcal{S}, \mathcal{A}, P, R, \gamma)$, where $\mathcal{S}$ is the state space, $\mathcal{A}$ is the action space, $P(s' \mid s, a)$ is state transition function from state $s \in \mathcal{S}$ to state $s' \in \mathcal{S}$, $R$ is the reward function

and $\gamma \in [0, 1)$ is the discount factor. Generally, the policy of an agent in an MDP is a mapping $\pi : \mathcal{S} \times \mathcal{A} \to [0, 1]$. An agent chooses actions $a \in \mathcal{A}$ according to a policy function $a \sim \pi(s)$, which updates the system state $s' \sim P(s, a)$ yielding a reward $r = R(s, a)$. In this paper, we denote a policy by $\pi_\theta$, where $\theta$ is the parameter of the policy function. The goal of the agent is to optimize the parameter $\theta$ for maximizing the expected accumulative rewards, $J(\pi_\theta) = \mathbb{E}_{\pi_\theta} \left[ \sum_{t=0}^{\infty} \gamma^t R(s_t, a_t) \right]$.

**State Discrepancy as Exploration Bonuses.** Addressing the sparse reward challenge prevalent in many environments necessitates augmenting the external reward function, $r^e$, with an intrinsic reward bonus, $b$, resulting in a combined reward: $r = r^e + b$. A number of intrinsic bonuses[2] that encourage exploration have been proposed. More recent methodologies employ an exploration bonus that varies based on the novelty of the current state relative to previously visited states, evaluated within different metric spaces, which can be generally defined as:

$$b = \text{SD} * \text{SF} = d(\Phi(s), \Phi(s')) * \lambda^{\text{SF}} \quad (1)$$

where $\text{SD}$ denotes the state discrepancy, $\text{SF}$ denotes the scaling factor, $d : \mathcal{S} \times \mathcal{S} \to \mathbb{R}$ represents the evaluation metric that quantifies the degree of difference or distance between two projected states, and $\Phi : \mathcal{S} \to \mathcal{Z}$ is the projection function which maps the state to a representation space $\mathcal{Z}$.

| Algorithm | Objective = State Discrepancy*Scaling Factor | Episodic | Approximation Gap | Scalability |
|---|---|---|---|---|
| RIDE [51] | $\|\phi(s') - \phi(s)\|_2 * \sqrt{N_{ep}(s')}^{-1}$ | ✔ | ✗ | ✗ |
| NovelD [69] | $\|r^{\text{RND}}(s') - \alpha \cdot r^{\text{RND}}(s)\|_+ * \mathbb{I}[N_{ep}(s') = 1]$ | ✔ | ✗ | ✗ |
| LIBERTY [63] | $[\gamma d_{inv}(s', s_0) - d_{inv}(s, s_0)] * \lambda$ | ✗ | ✔ | ✗ |
| EME (ours) | $d_E(s', s) * min\{max\{\zeta(r_s), 1\}, M\}$ | ✗ | ✗ | ✔ |

Table 1: Comparison of exploration methods using different measures (marked in blue) of state discrepancy as exploration bonus. $\phi = f(s)$ : features encoder; $N_{ep}(s)$ : episodic (pseudo) count of visits to state $s$; $r^{\text{RND}}(s)$: the RND [11] bonus; $\alpha$: normalized coefficient; $d_{inv}$: inverse dynamic bisimulation metric used in [63]; $s_0$: initial state; $\lambda$: the scaling hyper-parameter; $d_E$: our proposed effective metric for state discrepancy evaluation; $\zeta(r_s)$: variance of predictions from an ensemble of reward models: $M$: maximum reward scaling.

The comparison of recent metric-based exploration methods is summarized in Table 1 These bonuses reward high dissimilarity between adjacent states, encouraging exploration by leveraging measures such as the $L_2$ distance for state embeddings (as used by RIDE [51]) and the $L_1$ distance for the disparity in RND bonuses between adjacent states (as utilized by NovelD [69]). However, these approaches, grounded in $L_p$ norms and dependent on episodic state visit counts, face significant scalability challenges when exploring various environments. On the other hand, LIBERTY [63] proposes the inverse dynamic bisimulation metric to evaluate the state difference, which links state and value differences to enhance learning efficiency. Yet, the approximation inaccuracies inherent in bisimulation metric learning compromise theoretical assurances, breaking theoretical integrity of bisimulation metric. In the following sections, we detail the shortcomings of these methods.

## 3 Limitations of Existing Metric-based Exploration Bonus

### 3.1 Latent Vector Space under $L_p$ Norms

$L_p$ norms such as $L_1$ and $L_2$ norms are widely used in measuring the "distance" between states or embeddings. For example, considering a continuous map between the states of an MDP and a latent vector space: $\Phi : \mathcal{S} \to \mathcal{Z} \subseteq \mathbb{R}^n$. We can equip this vector space with a Euclidean norm to obtain a Euclidean metric space $(\mathcal{Z}, \|\cdot\|_2)$, and the state discrepancy is measured as $\text{SD} = \|\Phi(s) - \Phi(s')\|_2$.

**Reliance on Count-based Scaling Factor.** The state-difference based exploration bonus employed by RIDE [51] and Noveld [69] can be regarded as the difference between adjacent states in the $L_p$ norm space. The efficacy of the exploration bonus heavily lies in the episodic scaling factor ($\text{SF}$) measured by the count of visited states $N_{ep}(s)$ presented in Table 1. However, the count-based episodic bonus suffers from a fundamental limitation, which is similar to that faced by count-based approaches [5] in general: if each state is unique, then

Table 2: Mean success rate comparison (values below 0.5 are marked in red)

| Environment | Robosuite | MiniGrid | Habitat |
|---|---|---|---|
| RIDE | 0.52 | 0.93 | 0.19 |
| RIDE w/o EP | 0.05 ↓ | 0.00 ↓ | 0.00 ↓ |
| NovelD | 0.55 | 0.93 | 0.17 |
| NovelD w/o EP | 0.11 ↓ | 0.00 ↓ | 0.00 ↓ |
| LIBERTY | 0.73 | 0.00 | 0.13 |

---

[2]The detailed discussion of related work can be found in Appendix F

$N_{ep}(s_t)$ will always be 1 and the episodic bonus is no longer meaningful, which is the case for many real-world applications. For example, a household robot's state as recorded by its camera might include moving trees outside the window, clocks showing the time, or images on a television screen which are not relevant to its tasks, but nevertheless make each state unique. As shown in Table 2, without the episodic count term (denoted as "w/o EP"), the performance of RIDE and Noveld exhibits a significant decline in all different types of environments.

**Limited Expressiveness of Novelty.** As indicated in Table 2, the performance of RIDE and NovelD significantly degrades in environments with high-dimensional visual observations or continuous control tasks, where the success rate falls below 0.5. The significant decline highlights the inadequacies of utilizing state differences within vector spaces, computed using the $L_p$ norm, to effectively express novelty. Therefore, a more robust and effective metric is essential to accurately assess and characterize the novelty among visited states.

### 3.2 State Discrepancy within the Bisimulation Metric Space

The bisimulation metric [25, 26] defines a pseudometric $d : \mathcal{S} \times \mathcal{S} \to \mathbb{R}$ to measure the similarity between two states. The recently proposed variant, $\pi$-bisimulation metric [13], focuses on behaviors relative to a particular policy $\pi$, which consists of a reward difference term and a Wasserstein distance in dynamics models between states.

**Definition 1** ($\pi$-bisimulation metric). *Given a fixed policy $\pi$, the following $\pi$-bisimulation metric exists and is unique:*

$$d(s_i, s_j) = |\mathbb{E}_{a_i \sim \pi} r_{s_i}^{a_i} - \mathbb{E}_{a_j \sim \pi} r_{s_j}^{a_j}| + \gamma W_1(d)(P^\pi(\cdot \mid s_i), P^\pi(\cdot \mid s_j)) \tag{2}$$

*where $\mathbb{E}_{a_i \sim \pi} r_{s_i}^{a_i} = \mathbb{E}_{a_i \sim \pi(\cdot|s_i)} r(s_i, a_i), P^\pi(\cdot \mid s_i) = \mathbb{E}_{a \sim \pi(\cdot|s_i)} P(\cdot \mid s_i, a)$ and $W_1$ is the 1-Wasserstein distance.*

**Approximation Gap.** LIBERTY [63] proposes exploration bonuses based on state discrepancy within the inverse dynamic bisimulation metric space, which adds the difference between action output by the inverse dynamic model based on the bisimulation metric. The difference between states evaluated under the bisimulation metric is directly associated with the value difference (see Theorem 3.3 in [26] and Theorem 2 in [63] for detailed proof). The loss function of LIBERTY metric learning is:

$$\mathcal{L}(\phi) = \mathbb{E}\left[\left(d_{inv}(s_i, s_j; \phi) - |r_{s_i}^{a_i} - r_{s_j}^{a_j}| - \gamma W_2(P_{\bar{s}_i}^\pi(\eta), P_{\bar{s}_j}^\pi(\eta)) - \gamma |I_{\bar{s}_i}^{s_{i+1}}(\theta) - I_{\bar{s}_j}^{s_{j+1}}(\theta)|\right)^2\right] \tag{3}$$

where $\bar{s}$ denotes state with stop gradients, $P_{\bar{s}_i}^\pi(\eta)$ indicates probabilistic dynamics model parameterized with $\eta$ and $I_{\bar{s}_i}^{s_{i+1}}(\theta)$ is the inverse dynamic model parameterized with $\theta$. As the 1-Wasserstein distance is usually difficult to estimate, LIBERTY and prior methods [68] propose to use 2-Wasserstein distance $W_2$ to replace $W_1$, as $W_2$ has a convenient closed-form of a Gaussian distribution with respect to the $L_2$ distance.

**Proposition 1** (Relaxation Divergence). *Relaxing the $W_1$ metric to $W_2$ breaks the theoretical integrity of the inverse dynamic bisimulation metric [63] in scenarios where the transition dynamic model $P(s, a)$ or the policy $\pi$ is stochastic.*

Proof in Appendix B. Based on Proposition 1, we can see that the relaxation divergence of 1-Wasserstein distance between dynamic models can only be ignored when both the transition dynamics model and policy $\pi$ are deterministic. However, this assumption may be too strong to hold in practice.

Besides, the metric learning process encounters a theory-practice gap due to the relaxation of reward expectations. Specifically, computing the expected reward differences (first term of Equation (2)), $|\mathbb{E}_{a_i \sim \pi} r_{s_i}^{a_i} - \mathbb{E}_{a_j \sim \pi} r_{s_j}^{a_j}|$, proves to be computationally daunting and challenging to accurately estimate, even with sampling techniques. Following other bisimulation metric-based methods [13, 68], LIBERTY [63] shifts the expectation operator outside the absolute value of reward differences (second term of Equation (3)) and differences between action outputs predicted by inverse dynamic models, as in $\mathbb{E}_{a_i \sim \pi, a_j \sim \pi}[|r_{s_i}^{a_i} - r_{s_j}^{a_j}| + |I_{\bar{s}_i}^{s_{i+1}} - I_{\bar{s}_j}^{s_{j+1}}|]$, which facilitates more efficient sampling by avoiding the direct estimation of reward and action expectations.

**Proposition 2** (Shifted LIBERTY Distance). *The relaxation of expectation during learning process shifts the original LIBERTY distance and introduces a looser value difference bound.*

We extend the concept of the shifted MICo distance [14] introduced in [35, 16]. The proof is in appendix B. Based on Proposition 2, as the shifted LIBERTY distance has a looser value difference bound, it may be less relevant to the value function. As a result, the learned metric may not be able to capture the state similarity within bisimulation metric space accurately. , as the shifted LIBERTY distance has a looser value difference bound, it may be less relevant to the value function. As a result, the learned metric may not be able to capture the state similarity within bisimulation metric space accurately.

**Scalability Constraints.** As shown in Table 2, LIBERTY's performance significantly drops in hard exploration tasks within MiniGrid and in the realistic scenarios of the Habitat environment [60]. Specifically, in Habitat, agents must navigate through photorealistic simulations of actual indoor spaces. In these settings, the subtle variations between states diminish the impact of the exploration bonus, significantly lowering the efficiency of exploration. This limitation becomes even more apparent in the procedurally-generated MiniGrid environment, where the inherent stochasticity further impedes LIBERTY's performance. Those findings highlight the scalability challenges that bisimulation metric-based methods encounter in complex environments.

# 4  Effective Metric-based Exploration Bonus

In this section, we introduce the Effective Metric-based Exploration-bonus (EME), conceived to overcome the limitations identified in existing methods that use metric-based state discrepancy as exploration bonuses. Our development is driven by two primary objectives. Firstly, we aim to refine the metric learning process to achieve a more effective exploration without incurring the approximation gap, thereby upholding the theoretical integrity of value function bound. Secondly, we seek to obviate the reliance on scaling factor of episodic counts, ensuring effective exploration across different environments. To this end, we propose a novel metric that more precisely evaluates the behavioral similarity between states, establishing a more effective exploration strategy compared with previous approaches. Furthermore, we introduce an innovative approach that utilizes the variance between outputs of an ensemble of reward models in metric learning as a dynamically-adjusted scaling factor, which significantly enhances exploration efficiency, especially in hard exploration tasks, thereby improving scalability. The next two sections describe our method in detail.

## 4.1  The EME Metric

In order to avoid the approximation gap introduced by relaxation of Wasserstein distance and reward expectations mentioned in Proposition 1 and 2, along with the reliance on the episodic count, we eliminate the calculation of the Wasserstein distance, and propose the EME metric:

**Definition 2** (EME Distance Function). *Let* $\mathfrak{met}$ *be the space of bounded pseudo-metrics on state space* $\mathcal{S}$*, the EME metric* $d_E : \mathcal{S} \times \mathcal{S} \to \mathbb{R}$*, the EME distance function* $\mathcal{F}(d_E, \pi) : \mathfrak{met} \to \mathfrak{met}$ *is defined as:*

$$\mathcal{F}(d_E, \pi)(s_i, s_j) = |\mathbb{E}_{a_i \sim \pi} r_{s_i}^{a_i} - \mathbb{E}_{a_j \sim \pi} r_{s_j}^{a_j}| + \gamma \mathbb{E}_{\substack{a_i \sim \pi \\ a_j \sim \pi}} d_E(s_i', s_j') + \gamma D_{\mathrm{KL}}(\pi(\cdot|s_i)\|\pi(\cdot|s_j)) \quad (4)$$

*where* $D_{\mathrm{KL}}(\cdot\|\cdot)$ *represents the Kullback–Leibler (KL) divergence.*

**Behavioral Similarity Between States.** We strictly calculate the expectation of reward difference without relaxation. The EME metric measures the distribution distance between dynamics models by computing the distance between sampled subsequent states following representation learning method [14] to avoid the computation of the Wasserstein distance. Additionally, we integrate the Kullback–Leibler (KL) divergence between policy distributions to more robustly model the "**behavioral similarity**" between states, which is more effective and scalable across diverse environments with different observations, especially in addressing the critical "Noisy-TV" problem [11] during exploration. The "Noisy-TV" problem started as a thought experiment in [11] and is commonly discussed in exploration literature [51, 52, 42]. Imagine that an RL agent is rewarded with seeking the novel experience, a TV with unpredictable random noise outputs would be able to attract the agent's attention forever. The agent obtains new rewards from state discrepancy caused by noisy TV consistently, but it fails to make any meaningful action. To counter this, we utilize differences in policy distributions across states, promoting the exploration of diverse actions rather than repetitive behavior.

**Theorem 1.** *The EME distance function $\mathcal{F}(d_E, \pi) : \mathfrak{met} \to \mathfrak{met}$ has a unique fixed-point $\hat{d}_E$.*

Proof in Appendix B. Theorem 1 provides a convergence guarantee for the EME distance that by iterating $\mathcal{F}(d_E, \pi)$, distance will converge to the fixed-point $\hat{d}_E$.

**Theorem 2.** *(Guaranteed Value difference bound) Given the EME metric $d_E$, states $s_i$ and $s_j$, and a policy $\pi$, we have*

$$|V^\pi(s_i) - V^\pi(s_j)| \le d_E(s_i, s_j). \tag{5}$$

**Improved Exploration Efficiency.** The proof can be found in Appendix B. Theorem 2 demonstrates that the EME distance between states provides an upper bound on the difference in their state values. Let's recap the definition of one-step TD error: $\delta_t = r_{t+1} + \gamma V(s_{t+1}) - V(s_t)$. Intuitively, if there is a large value difference between states $s_t$ and $s_{t+1}$, the corresponding exploration bonus $b$ which is calculated using $d_E(s_t, s_{t+1})$ will also be large. Consequently, the total reward for the next step, $r_{t+1} = r^e + b$ (where $r^e$ represents the external environment reward), increases accordingly. The increase of reward leads to a larger TD error when there is a significant value difference between adjacent states, incentivizing the agent to prioritize transitions with large TD errors, which not only enhances the agent's exploration capabilities but also significantly boosts training efficiency.

## 4.2 Tractable Optimization of EME

Based on Equation (4), between any pair of states $s_i$ and $s_j$, we can define the loss function of the EME metric learning as:

$$\mathcal{L}(\phi) = \mathbb{E}\left[\left(d_E^\phi(s_i, s_j) - |\mathbb{E}_{a_i \sim \pi} r_{s_i}^{a_i} - \mathbb{E}_{a_j \sim \pi} r_{s_j}^{a_j}| - \gamma \mathbb{E}_{\substack{a_i \sim \pi \\ a_j \sim \pi}} d_E(s_i', s_j') - \gamma D_{\mathrm{KL}}(\pi(\cdot|s_i)\|\pi(\cdot|s_j))\right)^2\right] \tag{6}$$

where $d_E^\phi$ is the EME metric encoder parameterized by $\phi$. The last three terms in the loss function (6) can be regarded as the regression of target metric, which approximates the difference between rewards, distances of next states and distances of policy distributions, respectively. However, the reward expectations $\mathbb{E}_{a_i \sim \pi} r_{s_i}^{a_i}$ and $\mathbb{E}_{a_j \sim \pi} r_{s_j}^{a_j}$ are computationally intractable and also difficult to estimate even based on sampling [13]. And the relaxation of reward expectations $\mathbb{E}_{\substack{a_i \sim \pi \\ a_j \sim \pi}}|r_{s_i}^{a_i} - r_{s_j}^{a_j}|$ used by previous methods [63, 68, 13] will lead to the learned metric having a looser value difference bound than the original bisimulation metric as proved in Proposition 2. Inspired by metric-based representation learning methods [35, 16], we propose to use an ensemble of reward models to approximate the reward difference more accurately.

**Proposition 3.** *Let $r_s$ be a random variable over the action distribution defined by $p(r_s = r_s^a) = \pi(a|s)$, $var(r_{s_i})$ denote the variance of variable $r_{s_i}$, we can have:*

$$|\mathbb{E}_{a_i \sim \pi} r_{s_i}^{a_i} - \mathbb{E}_{a_j \sim \pi} r_{s_j}^{a_j}| = \sqrt{\mathbb{E}_{\substack{a_i \sim \pi \\ a_j \sim \pi}}\left[|r_{s_i}^{a_i} - r_{s_j}^{a_j}|^2\right] - var(r_{s_i}) - var(r_{s_j})} \tag{7}$$

*Proof.* (Sketch) The proof is based on [14, 16] by expanding $\mathbb{E}_{\substack{a_i \sim \pi \\ a_j \sim \pi}}[|r_{s_i}^{a_i} - r_{s_j}^{a_j}|^2] - |\mathbb{E}_{a_i \sim \pi} r_{s_i}^{a_i} - \mathbb{E}_{a_j \sim \pi} r_{s_j}^{a_j}|^2$. See detailed proof in Appendix B. $\square$

**Ensemble of Reward Models.** We train an ensemble of reward models $\{g(\eta_1), \ldots, g(\eta_k)\}$ to predict the reward from the state-action pairs sampling from the buffer $\mathcal{D}_\tau$: $g(s, a; \eta) : \mathcal{S} \times \mathcal{A} \to \mathcal{R}$ by minimizing the prediction error $\|g(s_t, a_t; \eta) - r_{t+1}\|^2$. Consequently, the variance of reward across the output of different models in the ensemble is defined as follows:

$$\mathrm{var}(r_{s_i}) \approx \zeta(r_{s_i}^{a_i}) = \mathbb{E}_{\substack{a_i \sim \pi(s_i) \\ (s_i, a_i) \sim \mathcal{D}_\tau}} \left\{\mathbb{E}_\eta\left[\|g(s_i, a_i, \eta) - \mathbb{E}_\eta[g(s_i, a_i, \eta)]\|_2^2\right]\right\} \tag{8}$$

Based on Equation (6), Equation (7) and Equation (8), the tractable EME loss without approximation gap can be defined as:

$$\mathcal{L}(\phi) = \mathbb{E}_{\mathcal{D}_\tau}[(d_E^\phi(s_i, s_j) - \sqrt{|r_{s_i}^{a_i} - r_{s_j}^{a_j}|^2 - \zeta(r_{s_i}^{a_i}) - \zeta(r_{s_j}^{a_j})} - \gamma\mathbb{E}_{\substack{a_i \sim \pi \\ a_j \sim \pi}} d_E^\phi(s_i', s_j')$$
$$- \gamma\mathbb{E}_{\pi(\cdot|s_i)}[\log\pi(\cdot|s_i) - \log\pi(\cdot|s_j)])^2] \tag{9}$$

**Diversity-Enhanced Scaling Factor.** The scaling factor of previous methods is either not scalable [51, 69, 2] or hand-crafted hyper-parameter [63], in light of this, we propose a scalable diversity-enhanced scaling factor for the intrinsic reward to further improve the efficacy of exploration. Let's recap the ensemble of reward models $\{g(\eta_1), \ldots, g(\eta_k)\}$ used in metric training (9). To maintain the diversity across the individual models $g(\eta)$, we initialize each model's parameters differently and train each of them on a subset of data randomly sampled with replacement. Each model in our ensemble is trained to predict the ground truth reward. Hence, the parts of reward obtained within the state space that have been well explored by the agent will have gathered enough data to train all models, resulting in a low variance of reward predictions from the models, when generalizing to unseen but similar parts of the reward from unvisited state-space, the areas which are novel and unexplored would still have high prediction error for all models as none of them are yet trained on such examples, resulting in higher variance of the reward prediction. Therefore, we use the variance of reward predictions as the scaling factor of intrinsic reward to encourage exploration on unvisited state space. With the maximum reward scaling $M$, the exploration bonus is defined as:

$$b_{t+1} = d_E(s_t, s_{t+1}) * min\{max\{\zeta(r_{s_t}), 1\}, M\} \tag{10}$$

As a result, the agent receives greater rewards when encountering novel states during training, which facilitates more effective exploration, the ablation study on scaling factor and visualization of exploration bonus is in Appendix C.5. See Algorithm 1 of Appendix D.2 for detailed description.

## 5 Experiments

The overall objective of our experiment is to evaluate the performance of EME in comparison to other baselines, we conduct comprehensive experiments on various settings of continuous control tasks, discrete-action hard exploration games, and real indoor environments[3] to assess the effectiveness and scalability of our algorithm. The implementation details can be found in Appendix D.

**Baselines.** We compare it against following competitive baseline methods. ICM [47]: a famous curiosity-driven method. RND [11]: intrinsic rewards are the prediction errors of the distillation network. E3B [32]: proposing count-based episodic bonuses under continuous state spaces. RIDE [51]: using state difference under latent space as exploration bonuses. NovelD [69]: using the state difference measured by RND bonus as intrinsic rewards. LIBERTY [63]: using the state discrepancy evaluated under the bisimulation metric space as shaping reward.

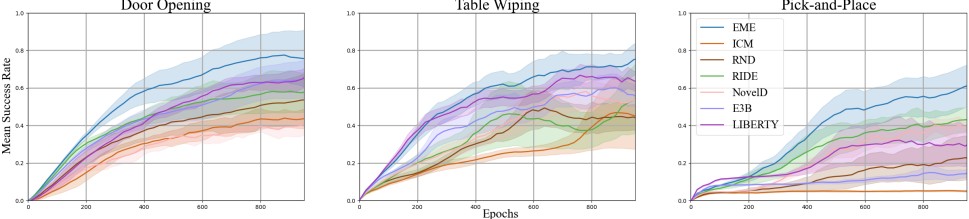

Figure 2: Results for various exploration tasks from Robosuite. The x-axis represents the number of epochs in training. The y-axis represents the mean success rate(standard deviations in shade).

### 5.1 Continuous Control

In our continuous control experiments conducted on the Robosuite platform [70], we assess the exploration capabilities of agents across various challenging tasks, which include Door Opening,

---

[3]Details of the environments are provided in Appendix D.1, and additional experiments are described in Appendix C.

Table 3: Average testing results of different Minigrid environments for EME and other baselines.

| | MRN7S8 | MRN7S8-NT | KCS4R3 | KCS4R3-NT | OM2D1h | OM2D1h-NT |
|---|---|---|---|---|---|---|
| ICM | $0.00 \pm 0.0$ | $0.00 \pm 0.0$ | $0.00 \pm 0.0$ | $0.00 \pm 0.0$ | $0.00 \pm 0.0$ | $0.00 \pm 0.0$ |
| RND | $0.00 \pm 0.0$ | $0.00 \pm 0.0$ | $0.00 \pm 0.0$ | $0.00 \pm 0.0$ | $0.95 \pm 0.001$ | $0.00 \pm 0.0$ |
| RIDE | $0.67 \pm 0.001$ | $0.65 \pm 0.001$ | $0.93 \pm 0.002$ | $0.81 \pm 0.001$ | $0.95 \pm 0.013$ | $0.89 \pm 0.015$ |
| NovelD | $0.67 \pm 0.001$ | $0.65 \pm 0.001$ | $\mathbf{0.93 \pm 0.003}$ | $0.00 \pm 0.0$ | $\mathbf{0.96 \pm 0.011}$ | $0.91 \pm 0.008$ |
| E3B | $0.61 \pm 0.003$ | $0.50 \pm 0.003$ | $0.91 \pm 0.007$ | $0.00 \pm 0.0$ | $0.91 \pm 0.012$ | $0.57 \pm 0.011$ |
| LIBERTY | $0.12 \pm 0.003$ | $0.00 \pm 0.0$ | $0.00 \pm 0.00$ | $0.00 \pm 0.0$ | $0.93 \pm 0.007$ | $0.51 \pm 0.005$ |
| EME | $\mathbf{0.67 \pm 0.011}$ | $\mathbf{0.65 \pm 0.003}$ | $0.91 \pm 0.011$ | $\mathbf{0.91 \pm 0.003}$ | $0.95 \pm 0.028$ | $\mathbf{0.95 \pm 0.011}$ |

Table Wiping, and Pick-and-Place. The detailed description of environments can be found in Appendix D.1.1. Each task represents a demanding context for robotic control, characterized by sparse rewards and significant exploratory challenges. The overall results are depicted in Figure 2, where our method consistently outperforms others, demonstrating its superiority in handling continuous control tasks. LIBERTY achieves the second-best performance in the Door Opening and Table Wiping tasks, underscoring the advantages of state discrepancy-based novelty within the bisimulation metric space. Conversely, episodic count-based methods such as RIDE, NovelD, and E3B lag behind, as episodic counts become less effective in environments with high-dimensional states. Furthermore, curiosity-driven approaches like ICM and RND struggle due to insufficient exploration. We also provide additional ablation studies on the ensemble size and max reward scaling in Appendix C.

**EME Combined with Feature Encoder.** The EME metric can be integrated with any state representation derived from various encoders $\Phi(\cdot)$, expressed as $d_E(\Phi(s_t), \Phi(s_{t+1}))$. The encoder may include the inverse dynamic model [47, 51, 63] which isolates environmental factors that do not affect the agent's behavior, the bisimulation-based encoder [68, 35, 14, 13], and random embeddings. The results can be found in Figure 8 in Appendix C.2, the performance of EME under the inverse dynamic encoder and bisimulation-based encoder exhibits a decline, because the exploration bonus under these encoders get close to zero quickly since their representations are very compact and every state looks more similar with the convergence of the encoder, which harms the efficacy of exploration. With respect to random embedding, the variance is larger, resulting in more unstable performance.

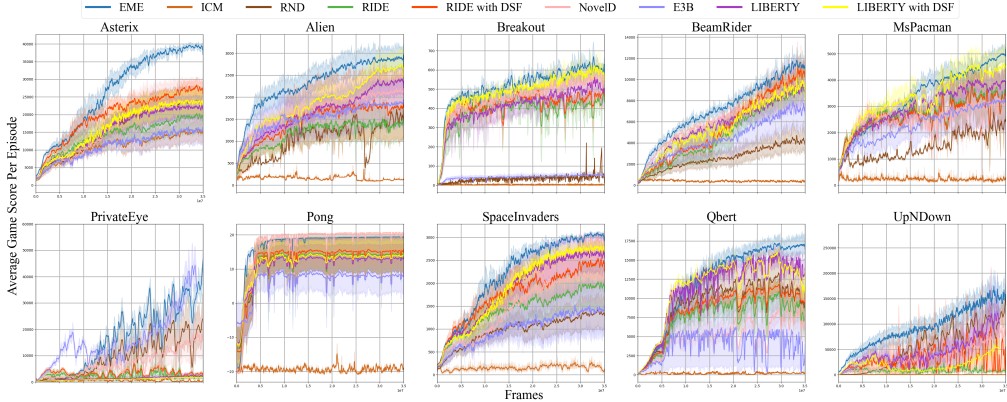

Figure 3: Comparison results for different hard exploration Atari games.

## 5.2 Discrete-action Games

**Atari Games.** To assess EME in scenarios involving pixel-based observations and discrete actions, we evaluate our method on the hard exploration games identified by [6, 69, 63] within the common Atari benchmark [8], including Asterix, Alien, Breakout, BeamRider, Mspacman, PrivateEye, Pong, SpaceInvaders, Qbert, and UpNDown. The results are presented in Figure 3. Our method consistently achieves similar or higher average returns compared to other baselines across all tested tasks. Notably, EME demonstrates the fastest convergence, particularly in Asterix and UpNDown, highlighting our method's superior learning efficiency. While LIBERTY and NovelD exhibit competitive performance in 6 games, their effectiveness significantly diminishes in the challenging games, PrivateEye and UpNDown. As for other methods like RIDE, E3B, ICM, and RND struggle across all tasks, primarily

due to inadequate exploration capabilities. To isolate the impact of our proposed EME metric, we also include the performance of RIDE and LIBERTY with Diversity-enhanced Scaling Factor, denoted as RIDE and LIBERTY with DSF. EME still achieves the best performance which demostrate the superiority of EME metric.

**MiniGrid Environments.** MiniGrid [17] presents a series of procedurally-generated, challenging environments. We focus on three settings from MiniGrid: Multi-Room (MR), Key Corridor (KC), and Obstructed Maze (OM). For example, MRN7S6-NT stands for MultiRoom-N7-S6-NoisyTV. Details on the specific environmental settings are available in Appendix D.1.3. Table 3 presents the testing performance of EME and all baseline methods across five different seeds. EME successfully solves all exploration tasks within MiniGrid, achieving the best performance in 4 out of 6 settings. While Multi-Room environments are relatively easy, all baselines except for ICM and RND, demonstrate competency. In the challenging Obstructed Maze environments, where obstructions block doors, our agent also excels by learning to remove these obstructions, further demonstrating the effectiveness and scalability of our approach.

**Noisy TV Problem.** In addition to standard MiniGrid tasks, we also tested the model's ability to deal with stochasticity in the environment by adding a manually-made Noisy TV setting introduced in [51], where some blocks change color at every time step. As illustrated in Table 3, EME maintains strong performance even under these conditions, underscoring the robustness of our method.

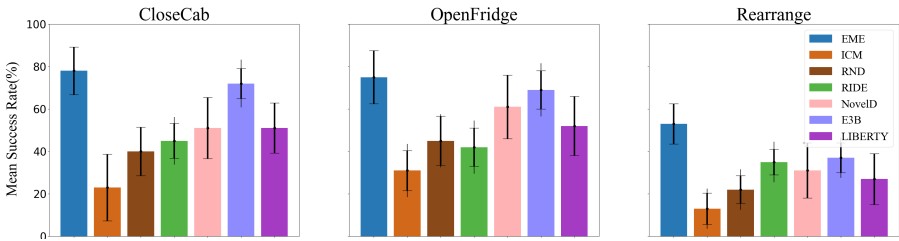

Figure 4: Results of exploration tasks on Habitat. Error bars represent std, deviations over 5 seeds.

## 5.3 Real-life Habitat Environment

To test our model's applicability to real-world environments, we investigate hard exploration tasks in Habitat [60]. Habitat is a platform for embodied AI research which provides an interface for agents to navigate and act in photorealistic (Figure 5) simulations of real indoor environments. Full details on the environmental setting can be found in Appendix D.1.2. We evaluate our methods on three embodied AI tasks of Habitat benchmark. As shown in Figure 4, EME consistently outperforms all baselines, demonstrating its superior scalability to high-dimensional visual-based observations and confirming its broad applicability. The performances of count-based methods like E3B, NovelD and RIDE are comparable, whereas curiosity-driven methods such as ICM and RND lag behind. The underperformance is primarily due to their inadequate exploration capabilities, which are particularly challenged by the complex and rich visual observations inherent in real-world settings.

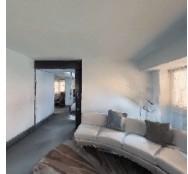

Figure 5: Habitat

**Reward-free Exploration.** we investigate reward-free exploration by evaluating the performance of all baselines using exploration bonus-only. Figure 9 displays the trajectories for all baselines on one of the test maps, clearly demonstrating that EME explores a significantly larger portion of the space compared to other methods. The full experiment results can be found in Appendix C.3.

## 6 Conclusion

In this work, we identify the limitations of existing state discrepancy-based exploration methods: the reliance on episodic count-based scaling factor and theory-practice approximation gap, which leads to limited scalability especially for hard exploration tasks within realistic environments. To address the issues, we propose the Effective Metric-based Exploration-bonus (EME), which addresses the inherent limitations by proposing a robust metric for state discrepancy evaluation backed by

comprehensive theoretical analysis. Furthermore, we propose the diversity-enhanced scaling factor integrated into the exploration bonus to enhance exploration effectiveness in particularly challenging scenarios. Extensive experiments on hard exploration tasks from continuous control, discrete-action games and realistic environments have demonstrated the effectiveness and scalability of our method.

## 7 Acknowledgements

This work was supported by National Science and Technology Major Project (2023ZD0121401), the Science and Technology Development Fund Macau SAR (0003/2023/RIC, 0052/2023/RIA1, 0031/2022/A, 001/2024/SKL for SKL-IOTSC), the Research Grant of University of Macau (MYRG2022-00252-FST), Shenzhen-Hong Kong-Macau Science and Technology Program Category C (SGDX20230821095159012), and Wuyi University Hong Kong and Macau joint Research Fund (2021WGALH14). This work was performed in part at SICC which is supported by SKL-IOTSC, University of Macau.

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

# A  Broader Impact Statement

This work introduces a reinforcement learning (RL) exploration method by introducing an effective metric-based exploration bonus, a versatile framework applicable to a wide array of decision-making problems, which can be applicable to diverse scenarios, including autonomous driving, household robotics, video gaming, online recommendation or advertisement optimization and etc. Similar to other RL exploration algorithms, our method is designed to facilitate the learning of a policy that maximizes a reward function defined by the designer. The implementation of such a policy, depending on the objectives set by the reward function designer, could lead to either positive or negative outcomes. Additionally, our approach significantly reduces the burden of reward tuning for researchers and algorithm engineers, and concurrently offers insights into the quality of the designed rewards through the shaping weights derived from our methods.

# B  Proofs

**Proposition 1** (Relaxation Divergence). *Relaxing the $W_1$ metric to $W_2$ breaks the theoretical integrity of the inverse dynamic bisimulation metric [63] in scenarios where the transition dynamic model $P(s, a)$ or the policy $\pi$ is stochastic.*

*Proof.* First, let's recall the proof of the existence of a unique fixed-point in the inverse dynamic bisimulation metric:

**Lemma 1** ([63]). *Given a fixed policy $\hat{\pi}$. Define $\mathcal{H}^{\hat{\pi}} : \mathfrak{met} \mapsto \mathfrak{met}$ by $\mathcal{H}(d, \hat{\pi})(s_i, s_j) = |r_{s_i}^{\hat{\pi}} - r_{s_j}^{\hat{\pi}}| + \gamma W(d)(\mathcal{P}_{s_i}^{\hat{\pi}}, \mathcal{P}_{s_j}^{\hat{\pi}}) + \|I^{\hat{\pi}}(\cdot \mid s_i, s_{i+1}) - I^{\hat{\pi}}(\cdot \mid s_j, s_{j+1})\|_1$, then $\mathcal{H}^{\hat{\pi}}$ has a least fixed point $d^{\hat{\pi}}$.*

*Proof.* This proof mimics the proof of Theorem $4.5$ from [24]. We make use of the same pointwise ordering on $\mathfrak{met}$: $d \leq d'$ iff $d(s, t) \leq d'(s, t)$ for all $s, t \in \mathcal{S}$, which gives us an $\omega$-cpo with bottom $\bot$, which is the everywhere-zero metric. Since Lemma $4.4$ from [24] (Wasserstein metric $W$ is continuous) also applies in our definition, it only remains to show that $\mathcal{H}(d, \hat{\pi})$ is continuous:

$$
\begin{aligned}
\mathcal{H}^{\hat{\pi}}\left(\bigsqcup_{n\in\mathbb{N}}\{x_n\}\right)(s_i, s_j) &= |r_{s_i}^{\hat{\pi}} - r_{s_j}^{\hat{\pi}}| + \gamma W\left(\bigsqcup_{n\in\mathbb{N}}\{x_n\}\right)(\mathcal{P}_{s_i}^{\hat{\pi}}, \mathcal{P}_{s_j}^{\hat{\pi}}) \\
&\quad + \|I^{\hat{\pi}}(\cdot \mid s_i, s_{i+1}) - I^{\hat{\pi}}(\cdot \mid s_j, s_{j+1})\|_1 \\
&= |r_{s_i}^{\hat{\pi}} - r_{s_j}^{\hat{\pi}}| + \gamma \sup_{n\in\mathbb{N}} W\left(x_n\right)(\mathcal{P}_{s_i}^{\hat{\pi}}, \mathcal{P}_{s_j}^{\hat{\pi}}) \\
&\quad + \|I^{\hat{\pi}}(\cdot \mid s_i, s_{i+1}) - I^{\hat{\pi}}(\cdot \mid s_j, s_{j+1})\|_1 \\
&\quad \text{by continuity of } W \\
&= \sup_{n\in\mathbb{N}}(|r_{s_i}^{\hat{\pi}} - r_{s_j}^{\hat{\pi}}| + \gamma W\left(x_n\right)(\mathcal{P}_{s_i}^{\hat{\pi}}, \mathcal{P}_{s_j}^{\hat{\pi}}) \\
&\quad + \|I^{\hat{\pi}}(\cdot \mid s_i, s_{i+1}) - I^{\hat{\pi}}(\cdot \mid s_j, s_{j+1})\|_1) \\
&= \sup_{n\in\mathbb{N}}\left\{\mathcal{H}^{\hat{\pi}}\left(x_n\right)(s_i, s_j)\right\} \\
&= \left(\bigsqcup_{n\in\mathbb{N}}\left\{\mathcal{H}^{\hat{\pi}}\left(x_n\right)\right\}\right)(s_i, s_j)
\end{aligned}
\tag{11}
$$

The rest of the proof follows in the same way as in [24]. $\qquad\square$

In equation (11), the existence of a unique fixed-point in the bisimulation metric requires the continuity and monotonicity of $W_1$ with respect to $d$. The properties of continuity and monotonicity do not hold with $W_2$. Therefore there is no more guarantee about the fixed-point existence in LIBERTY and other approximation-based methods [68] except that both the dynamics model and the policy $\pi$ are deterministic, in which case $W_2(d)$ degenerates to $d$ [24] and Banach's fixed-point exists [35]. As for scenarios when the transition model and policy are stochastic, the theoretical guarantee of LIBERTY is broken. $\qquad\square$

**Proposition 2** (Shifted LIBERTY Distance). *The relaxation of expectation during learning process shifts the original LIBERTY distance and introduces a looser value difference bound.*

*Proof.* The proof is based on top of [14, 16].

**Lemma 2.**

$$\mathbb{E}_{a_i\sim\pi,a_j\sim\pi}[|r_{s_i}^{a_i}-r_{s_j}^{a_j}|+|I_{\bar{s}_i}^{s_{i+1}}-I_{\bar{s}_j}^{s_{j+1}}|] \geq |\mathbb{E}_{a_i\sim\pi}r_{s_i}^{a_i}-\mathbb{E}_{a_j\sim\pi}r_{s_j}^{a_j}|+|\mathbb{E}_{a_i\sim\pi}I_{\bar{s}_i}^{s_{i+1}}-\mathbb{E}_{a_j\sim\pi}I_{\bar{s}_j}^{s_{j+1}}|$$

(12)

*Proof.* Since $r_{s_i}^{a_i}$ and $r_{s_j}^{a_j}$ are rewards which are scalars, we have $|r_{s_i}^{a_i}-r_{s_j}^{a_j}| \geq r_{s_i}^{a_i}-r_{s_j}^{a_j}$ and $|r_{s_j}^{a_j}-r_{s_i}^{a_i}| \geq r_{s_j}^{a_j}-r_{s_i}^{a_i}$. By taking the expectation over $a_i$ and $a_j$, we can have

$$\mathbb{E}_{a_i\sim\pi,a_j\sim\pi}|r_{s_i}^{a_i}-r_{s_j}^{a_j}| \geq \mathbb{E}_{a_i\sim\pi,a_j\sim\pi}[r_{s_i}^{a_i}-r_{s_j}^{a_j}]$$

(13)

and

$$\mathbb{E}_{a_i\sim\pi,a_j\sim\pi}|r_{s_j}^{a_j}-r_{s_i}^{a_i}| \geq \mathbb{E}_{a_i\sim\pi,a_j\sim\pi}[r_{s_j}^{a_j}-r_{s_i}^{a_i}]$$

(14)

By combining Equation (13) and Equation (14), we can have:

$$\mathbb{E}_{a_i\sim\pi,a_j\sim\pi}|r_{s_i}^{a_i}-r_{s_j}^{a_j}| \geq \mathbb{E}_{a_i\sim\pi,a_j\sim\pi}[r_{s_i}^{a_i}-r_{s_j}^{a_j}] = |\mathbb{E}_{a_i\sim\pi}r_{s_i}^{a_i}-\mathbb{E}_{a_j\sim\pi}r_{s_j}^{a_j}|$$

(15)

Similarly, we can have:

$$\mathbb{E}_{a_i\sim\pi,a_j\sim\pi}|I_{\bar{s}_i}^{s_{i+1}}-I_{\bar{s}_j}^{s_{i+j}}| \geq \mathbb{E}_{a_i\sim\pi,a_j\sim\pi}[I_{\bar{s}_i}^{s_{i+1}}-I_{\bar{s}_j}^{s_{i+j}}] = |\mathbb{E}_{a_i\sim\pi}I_{\bar{s}_i}^{s_{i+1}}-\mathbb{E}_{a_j\sim\pi}I_{\bar{s}_j}^{s_{j+1}}|$$

(16)

So we can get

$$\mathbb{E}_{a_i\sim\pi,a_j\sim\pi}[|r_{s_i}^{a_i}-r_{s_j}^{a_j}|+|I_{\bar{s}_i}^{s_{i+1}}-I_{\bar{s}_j}^{s_{j+1}}|] \geq |\mathbb{E}_{a_i\sim\pi}r_{s_i}^{a_i}-\mathbb{E}_{a_j\sim\pi}r_{s_j}^{a_j}|+|\mathbb{E}_{a_i\sim\pi}I_{\bar{s}_i}^{s_{i+1}}-\mathbb{E}_{a_j\sim\pi}I_{\bar{s}_j}^{s_{j+1}}|$$

(17)

□

**Definition 3** (Shifted LIBERTY Distance). *The shifted LIBERTY distance function $\hat{\mathcal{H}}^\pi$ is defined as*

$$\hat{\mathcal{H}}^\pi(d)(s_i,s_j) = \mathbb{E}_{\substack{a_i\sim\pi\\a_j\sim\pi}}|r_{s_i}^{a_i}-r_{s_j}^{a_j}| - \gamma W_1(P_{\bar{s}_i}^\pi, P_{\bar{s}_j}^\pi) - \gamma|I_{\bar{s}_i}^{s_{i+1}}-I_{\bar{s}_j}^{s_{j+1}}|$$

(18)

Define the MDP for RL by a tuple $\langle\mathcal{S},\mathcal{A},\mathcal{R},\mathcal{P},\gamma\rangle$. We consider a lifted MDP constructed by a tuple $\langle\hat{\mathcal{S}},\hat{\mathcal{A}},\hat{\mathcal{R}},\hat{\mathcal{P}},\gamma\rangle$, where state space $\hat{\mathcal{S}} = \mathcal{S}\times\mathcal{S}$, action space $\hat{\mathcal{A}} = \mathcal{A}\times\mathcal{A}$, transition distribution $\hat{\mathcal{P}}_{(s_i,s_j)}^{(a_i,a_j)} = P_{s_i}^{a_i}P_{s_j}^{a_j}$, and reward function $\hat{\mathcal{R}}((s_i,s_j)) = |\mathbb{E}_{a_i\sim\pi}r_{s_i}^{a_i}-\mathbb{E}_{a_j\sim\pi}r_{s_j}^{a_j}|$. The Bellman operator $T^\pi$ under policy $\pi(a_i,a_j|s_i,s_j) = \pi(a_i|s_i)\pi(a_j|s_j)$ is:

$$
\begin{aligned}
T^\pi(d^\pi)((s_i,s_j)) &= \sum_{(a_i,a_j)}\pi(a_i,a_j|s_i,s_j)\sum_{(s_i',s_j')}\hat{\mathcal{P}}_{(s_i,s_j)}^{(a_i,a_j)}(s_i',s_j')\left[\hat{\mathcal{R}}((s_i,s_j))+\gamma|I_{s_i}^{s_i'}-I_{s_j}^{s_j'}|\right]\\
&= \left|\mathbb{E}_{a_i\sim\pi}r_{s_i}^{a_i}-\mathbb{E}_{a_j\sim\pi}r_{s_j}^{a_j}\right|+\gamma W(P_{\bar{s}_i}^\pi,P_{\bar{s}_j}^\pi)+\gamma|I_{s_i}^{s_i'}-I_{s_j}^{s_j'}|\\
&= \mathcal{H}^\pi(d^\pi)((s_i,s_j)).
\end{aligned}
$$

(19)

So we can construct the lifted MDP of shift LIBERTY by a tuple $\langle\hat{\mathcal{S}},\hat{\mathcal{A}},\tilde{\mathcal{R}},\hat{\mathcal{P}},\gamma\rangle$ which is the same as the lifted MDP in Definition 3 except for the reward function $\tilde{\mathcal{R}}((s_i,s_j),(a_i,a_j)) = |r_{s_i}^{a_i}-r_{s_j}^{a_j}|$ and shifted metric $\tilde{d}$. The Bellman operator $\tilde{T}^{\hat{\pi}}$ under policy $\hat{\pi}$ is:

$$
\begin{aligned}
\tilde{T}^{\hat{\pi}}(\tilde{d}^{\hat{\pi}})((s_i,s_j)) &= \sum_{(a_i,a_j)}\hat{\pi}(a_i,a_j|s_i,s_j)\sum_{(s_i',s_j')}\hat{\mathcal{P}}_{(s_i,s_j)}^{(a_i,a_j)}(s_i',s_j')\left[\tilde{\mathcal{R}}((s_i,s_j),(a_i,a_j))+\gamma|I_{s_i}^{s_i'}-I_{s_j}^{s_j'}|\right]\\
&= \mathcal{H}^{\hat{\pi}}(\tilde{d}^{\hat{\pi}})((s_i,s_j))
\end{aligned}
$$

(20)

As proven in LIBERTY [63], for the inverse dynamic bisimulation metric $d$, we can get

$$|V^\pi(s_i)-V^\pi(s_j)| \leq d^\pi(s_i,s_j)$$

(21)

As proved in [63, 24, 27], $d^\pi$ is the value function of lifted MDP $\langle \hat{\mathcal{S}}, \hat{\mathcal{A}}, \hat{\mathcal{R}}, \hat{\mathcal{P}}, \gamma \rangle$. $d^\pi$ can be expanded as the sum of discounted future rewards,

$$d^\pi(s_i, s_j) = \mathbb{E}_{\hat{\pi}}\left[\sum_t \gamma^t \left(\mathbb{E}_{a_i^{(t)} \sim \pi} r_{s_i^{(t)}}^{a_i^{(t)}} - \mathbb{E}_{a_j^{(t)} \sim \pi} r_{s_j^{(t)}}^{a_j^{(t)}}\right) \Bigg| s_i^{(0)} = s_i, s_j^{(0)} = s_j\right]. \qquad (22)$$

$\tilde{d}^\pi$ can be expanded as,

$$\tilde{d}^\pi(s_i, s_j) = \mathbb{E}_{\hat{\pi}}\left[\sum_t \gamma^t \left(\left|\mathbb{E}_{a_i^{(t)} \sim \pi} r_{s_i^{(t)}}^{a_i^{(t)}} - \mathbb{E}_{a_j^{(t)} \sim \pi} r_{s_j^{(t)}}^{a_j^{(t)}}\right|\right) \Bigg| s_i^{(0)} = s_i, s_j^{(0)} = s_j\right]. \qquad (23)$$

with $s_i^{(0)} = s_i, s_j^{(0)} = s_j$, The difference between $d$ and $\tilde{d}$ can be regarded as

$$\tilde{d}^\pi(s_i, s_j) - d^\pi(s_i, s_j) = \mathbb{E}_{\hat{\pi}}\left[\sum_t \gamma^t \left(\left|\mathbb{E}_{a_i^{(t)} \sim \pi} r_{s_i^{(t)}}^{a_i^{(t)}} - \mathbb{E}_{a_j^{(t)} \sim \pi} r_{s_j^{(t)}}^{a_j^{(t)}}\right| - \left|\mathbb{E}_{a_i^{(t)} \sim \pi} r_{s_i^{(t)}}^{a_i^{(t)}} - \mathbb{E}_{a_j^{(t)} \sim \pi} r_{s_j^{(t)}}^{a_j^{(t)}}\right|\right)\right]$$

$$= \mathbb{E}_{\hat{\pi}}\left[\sum_t \gamma^t \hat{\mathcal{R}}_\Delta((s_i^{(t)}, s_j^{(t)}), (a_i^{(t)}, a_j^{(t)}))\right] \qquad (24)$$

As proved in Lemma 2, we can get:

$$\tilde{d}^\pi(s_i, s_j) - d^\pi(s_i, s_j) \geq 0 \qquad (25)$$

Combined with Equation (21), we can have:

$$|V^\pi(s_i) - V^\pi(s_j)| \leq d^\pi(s_i, s_j) \leq \tilde{d}^\pi(s_i, s_j) \qquad (26)$$

So the shifted LIBERTY distance introduces a looser value bound. $\qquad\square$

**Proposition 3.** *Let $r_s$ be a random variable over the action distribution defined by $p(r_s = r_s^a) = \pi(a|s)$, $var(r_{s_i})$ denote the variance of variable $r_{s_i}$, we can have:*

$$|\mathbb{E}_{a_i \sim \pi} r_{s_i}^{a_i} - \mathbb{E}_{a_j \sim \pi} r_{s_j}^{a_j}| = \sqrt{\mathbb{E}_{\substack{a_i \sim \pi \\ a_j \sim \pi}}\left[|r_{s_i}^{a_i} - r_{s_j}^{a_j}|^2\right] - var(r_{s_i}) - var(r_{s_j})} \qquad (27)$$

*Proof.* The proof is based on [14, 16] by expanding the difference between $\mathbb{E}_{a_i \sim \pi, a_j \sim \pi}\left[(r_{s_i}^{a_i} - r_{s_j}^{a_j})^2\right]$ and $\left(\mathbb{E}_{a_i \sim \pi} r_{s_i}^{a_i} - \mathbb{E}_{a_j \sim \pi} r_{s_j}^{a_j}\right)^2$.

$$\mathbb{E}_{a_i \sim \pi, a_j \sim \pi}\left[\left(r_{s_i}^{a_i} - r_{s_j}^{a_j}\right)^2\right] - \left(\mathbb{E}_{a_i \sim \pi} r_{s_i}^{a_i} - \mathbb{E}_{a_j \sim \pi} r_{s_j}^{a_j}\right)^2$$

$$= \mathbb{E}_{a_i \sim \pi}\left[\left(r_{s_i}^{a_i}\right)^2\right] + \mathbb{E}_{a_j \sim \pi}\left[\left(r_{s_j}^{a_j}\right)^2\right] - 2\mathbb{E}_{a_i \sim \pi, a_j \sim \pi}\left[r_{s_i}^{a_i} r_{s_j}^{a_j}\right]$$

$$- \left(\left(\mathbb{E}_{a_i \sim \pi} r_{s_i}^{a_i}\right)^2 + \left(\mathbb{E}_{a_j \sim \pi} r_{s_j}^{a_j}\right)^2 - 2\mathbb{E}_{a_i \sim \pi} r_{s_i}^{a_i} \mathbb{E}_{a_j \sim \pi} r_{s_j}^{a_j}\right) \qquad (28)$$

$$= \mathbb{E}_{a_i \sim \pi}\left[\left(r_{s_i}^{a_i}\right)^2\right] - \left(\mathbb{E}_{a_i \sim \pi} r_{s_i}^{a_i}\right)^2 + \mathbb{E}_{a_j \sim \pi}\left[\left(r_{s_j}^{a_j}\right)^2\right] - \left(\mathbb{E}_{a_j \sim \pi} r_{s_j}^{a_j}\right)^2$$

$$- 2\left(\mathbb{E}_{a_i \sim \pi, a_j \sim \pi}\left[r_{s_i}^{a_i} r_{s_j}^{a_j}\right] - \mathbb{E}_{a_i \sim \pi} r_{s_i}^{a_i} \mathbb{E}_{a_j \sim \pi} r_{s_j}^{a_j}\right)$$

$$= var[r_{s_i}] + var(r_{s_j}) - 2cov(r_{s_i}, r_{s_j})$$

$r_{s_i}$ and $r_{s_j}$ are independent variables, we can get $cov(r_{s_i}, r_{s_j}) = 0$, so we can have

$$|\mathbb{E}_{a_i \sim \pi} r_{s_i}^{a_i} - \mathbb{E}_{a_j \sim \pi} r_{s_j}^{a_j}| = \sqrt{\mathbb{E}_{\substack{a_i \sim \pi \\ a_j \sim \pi}}\left[|r_{s_i}^{a_i} - r_{s_j}^{a_j}|^2\right] - var(r_{s_i}) - var(r_{s_j})} \qquad (29)$$

$\qquad\square$

**Theorem 1.** *The EME distance function $\mathcal{F}(d_E, \pi) : \mathfrak{met} \rightarrow \mathfrak{met}$ has a unique fixed-point $\hat{d}_E$.*

*Proof.* Let $d_E, d'_E \in \mathbb{M}$. We have

$$|\mathcal{F}(d_E)(s_i, s_j) - \mathcal{F}(d'_E)(s_i, s_j)| = \left| \gamma \sum_{a_i, a_j} \pi(a_i \mid s_i) \pi(a_j \mid s_j)(d_E - d'_E)(s'_i, s'_j) \right| \tag{30}$$

$$\leq \gamma \|d_E - d'_E\|_\infty$$

Therefore, $\mathcal{F}$ is a contraction mapping w.r.t. the $L_\infty$ norm and there exists a unique fixed-point for $\mathcal{F}$ due to Banach's fixed-point theorem. This completes the proof. $\qquad\square$

**Theorem 2.** *(Guaranteed Value difference bound) Given the EME metric $d_E$, states $s_i$ and $s_j$, and a policy $\pi$, we have*

$$|V^\pi(s_i) - V^\pi(s_j)| \leq d_E(s_i, s_j). \tag{31}$$

*Proof.* The proof mimics [14]. We follow the assumption that $\sum_{s'} P_s^a(s') V^\pi(s') = V^\pi(\mathbb{E}_{s' \sim P_s^a}[s'])$. We will first show that if $\forall s_i, s_j \in \mathcal{S}, |V^\pi(s_i) - V^\pi(s_j)| \leq d(s_i, s_j)$, then $|V^\pi(s_i) - V^\pi(s_j)| \leq \mathcal{F}(d_E, \pi)(s_i, s_j)$. Suppose $|V^\pi(s_i) - V^\pi(s_j)| \leq d(s_i, s_j)$ holds, we can have:

$$
\begin{aligned}
&V^\pi(s_i) - V^\pi(s_j) \\
&= \mathbb{E}_{a_i \sim \pi} r_{s_i}^{a_i} - \mathbb{E}_{a_j \sim \pi} r_{s_j}^{a_j} + \sum_{a_i} \pi(a_i|s_i) \sum_{s'_i} P_{s_i}^{a_i}(s'_i) V^\pi(s'_i) - \sum_{a_j} \pi(a_j|s_j) \sum_{s'_j} P_{s_j}^{a_j}(s'_j) V^\pi(s'_j) \\
&\leq \left| \mathbb{E}_{a_i \sim \pi} r_{s_i}^{a_i} - \mathbb{E}_{a_j \sim \pi} r_{s_j}^{a_j} \right| + \mathbb{E}_{a_i \sim \pi}(\sum_{s'_i} P_{s_i}^{a_i}(s'_i) V^\pi(s'_i)) - \mathbb{E}_{a_j \sim \pi}(\sum_{s'_j} P_{s_j}^{a_j}(s'_j) V^\pi(s'_j)) \\
&\leq \left| \mathbb{E}_{a_i \sim \pi} r_{s_i}^{a_i} - \mathbb{E}_{a_j \sim \pi} r_{s_j}^{a_j} \right| + \mathbb{E}_{a_i \sim \pi} \left( V^\pi \left( \mathbb{E}_{s' \sim P_{s_i}^{a_i}}[s'_i] \right) - V^\pi \left( \mathbb{E}_{s' \sim P_{s_j}^{a_j}}[s'_j] \right) \right) \\
&\leq \left| \mathbb{E}_{a_i \sim \pi} r_{s_i}^{a_i} - \mathbb{E}_{a_j \sim \pi} r_{s_j}^{a_j} \right| + \mathbb{E}_{a_i \sim \pi} d \left( \mathbb{E}_{s' \sim P_{s_i}^{a_i}}[s'_i], \mathbb{E}_{s' \sim P_{s_j}^{a_j}}[s'_j] \right) \\
&\leq \left| \mathbb{E}_{a_i \sim \pi} r_{s_i}^{a_i} - \mathbb{E}_{a_j \sim \pi} r_{s_j}^{a_j} \right| + \gamma \mathbb{E}_{a_i \sim \pi} d \left( s'_i, s'_j \right) + \gamma \mathbb{E}_{\pi(\cdot|s_i)} [\log \pi(\cdot|s_i) - \log \pi(\cdot|s_j)] \\
&\leq \mathcal{F}(d, \pi)(s_i, s_j))
\end{aligned}
\tag{32}
$$

Similarly, $V^\pi(s_j) - V^\pi(s_i) \leq \mathcal{F}(d, \pi(s_j, s_i))$, so we can have

$$|V^\pi(s_i) - V^\pi(s_j)| \leq \mathcal{F}(d, \pi)(s_i, s_j)) \tag{33}$$

Assuming that we have an initial distance $d_0$ which $|V^\pi(s_i) - V^\pi(s_j)| \leq d_0(s_i, s_j)$, and base on Theorem 1, $\mathcal{F}(d, \pi)$ is contraction mapping on $d$. By repeatedly applying $\mathcal{F}(d, \pi)$ on $d$, $d$ will eventually converge to the fixed-point $d_E$, the fixed point $d_E$ satisfies:

$$|V^\pi(s_i) - V^\pi(s_j)| \leq d_E(s_i, s_j)) \tag{34}$$

$$\square$$

## C  Additional Experiments

### C.1  Additional Ablation Experiments

To isolate the impact of different hyper-parameters, we carry additional experiments on the ensemble size of reward models and the max reward scaling $M$. First, $M$ sets an upper limit on the bonus. We set $M = 10$ as the default setting. As we can see from Figure 6, if $M = 1$, the scaling factor is fixed to 1, resulting in a significant performance decline. Higher $M$ encourages more extensive exploration. The performance with $M = 5$ slightly lags behind the default setting. The performance with $M = 20$ and $M = 40$ is comparable and almost the same, indicating that the performance stabilizes as $M$ increases. Practically, the value of $M$ can be adjusted depending on the specific task and environment to determine the intensity of exploration.

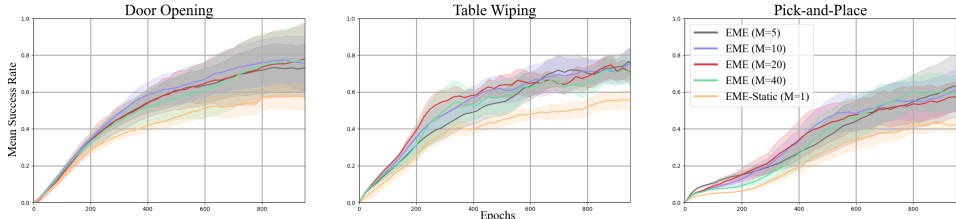

Figure 6: Ablation study on the max reward scaling $M$ ($M = 1, 5, 10, 20, 40$).

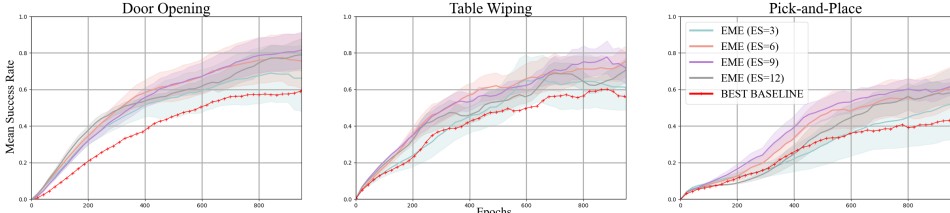

Figure 7: Ablation study on the ensemble size (ES = 3, 6, 9, 12) and the other baseline with best performance.

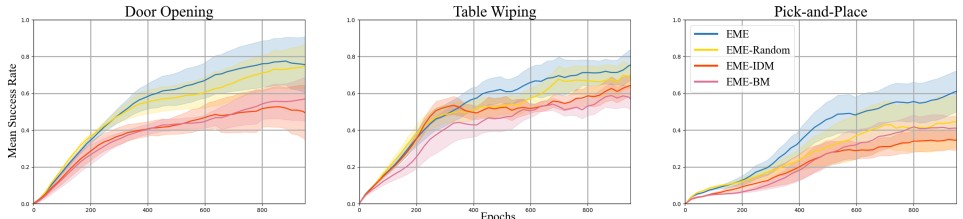

Figure 8: Results for EME and its variants combined with different feature encoders. The x-axis represents the number of epochs in training. The y-axis represents the mean success rate(standard deviations in shade).

As shown in Figure 7, we observe that as the ensemble size (ES) increases, the performance of ES = 6 and ES = 9 surpasses that of ES = 3. The performance of ES = 6 and ES = 9 is comparable, with no significant difference. However, when the size is further increased to ES = 12, there is a decline in performance, particularly in the Table Wiping and Pick and Place tasks. The performance change can be analyzed from two perspectives. First, the number of reward models is related to the accuracy of the reward variance prediction used to calculate the loss function. As the ensemble size increases, the overall prediction error decreases because the models can average out individual errors more effectively. This leads to a more accurate approximation of the variance $\zeta(s_t)$. Second, with a larger ensemble size, the variance of reward predictions decreases, resulting in a lower scale of the exploration bonus. Thus, there is a balance or trade-off between estimation accuracy and exploration, explaining why the performance of ES = 12 lags behind ES = 9. The optimal number of ensemble models may vary depending on the specific tasks and environments. Regarding computational cost, it increases with the ensemble size. Therefore, we set ES = 6 as our default setting, where the performance is nearly the same as ES = 9 but with lower computational overhead. It is also noteworthy that even with an ensemble size of 3, EME still outperforms the best baseline methods, further demonstrating the robustness of EME.

## C.2   EME Combined with Feature Encoder

Our method EME can be integrated with any state representation derived from different state encoders $\Phi(\cdot)$, expressed as $d_E(\Phi(s_t), \Phi(s_{t+1}))$. The encoder may include the inverse dynamic model [47, 51, 63] which isolates environmental factors that do not affect the agent's behavior, the bisimulation-based encoder [68, 35, 14, 13] which learns a compact state representation grouped by the bisimulation metric, and random embeddings captured by random embedded states. We denote the EME variants as follows: EME with an inverse dynamic model-based encoder is referred to as EME-IDM; with a bisimulation metric-based encoder as EME-BM; and with a random embedding as EME-Random. The results can be found in Figure 8. EME's performance declines due to the rapid reduction of the

exploration bonus. The performance of EME under the inverse dynamic encoder and bisimulation-based encoder exhibits a decline, because the exploration bonus under these encoders gets close to zero quickly since their representations are very compact and every state looks more similar with the convergence of the encoder, which harms the efficacy of exploration. With respect to random embedding, the variance is larger, resulting in more unstable performance.

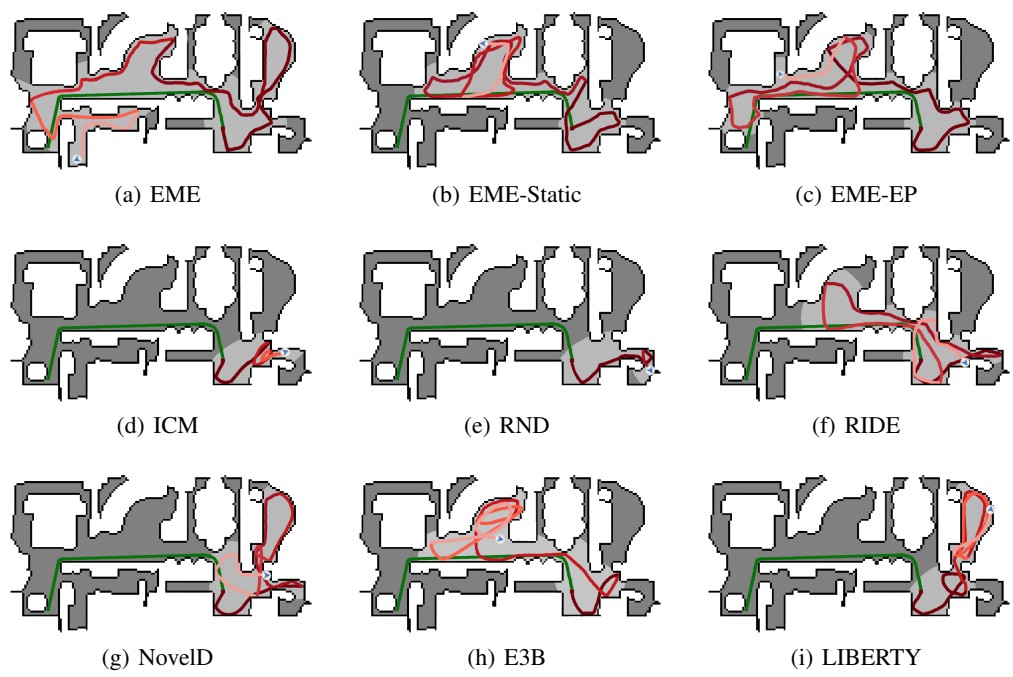

Figure 9: Trajectories of policies trained with different exploration algorithms on the Habitat environment. Our method EME reveals the largest portion of the map than other methods.

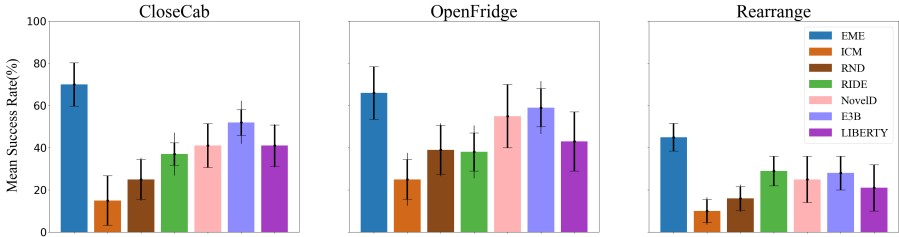

Figure 10: Results of reward-free exploration tasks on Habitat. Error bars represent std, deviations over 5 seeds.

### C.3 Reward-free Exploration

We examine reward-free exploration by assessing the performance of all baselines within Habitat using only exploration bonuses. During training, agents are initialized in varied environments for each episode and are subsequently tested on a set of environments not encountered during training. Figure 9 displays the trajectories of all baselines on one of the test maps, highlighting that EME explores a substantial portion of the space, unlike other methods. Specifically, ICM and RND agents remain confined to a single room, while RIDE, LIBERTY, and NovelD agents explore two rooms before becoming stuck, distant from the goal. E3B exhibits the second-best exploration coverage of the map. Quantitative results from other reward-free tasks are shown in Figure 10, where EME consistently demonstrates robust performance and achieves the best outcomes. In contrast, other methods experience significant declines, underscoring the efficacy of our approach in a reward-free setting.

Table 4: Table of quantitative results comparison between EME and other baseline methods in different environments of Mujoco with the delayed reward setting. The best and the runner-up results are (**bold**) and (underline)

| Methods | Delay = 10 | | | | | |
|---|---|---|---|---|---|---|
| | HalfCheetah | Hopper | Walker2d | Ant | Humanoid | Swimmer |
| ICM | $1374 \pm 368$ | $1258 \pm 325$ | $1127 \pm 225$ | $-105 \pm 43$ | $462 \pm 54$ | $27 \pm 11$ |
| RND | $1694 \pm 495$ | $1976 \pm 458$ | $1405 \pm 262$ | $143 \pm 17$ | $532 \pm 29$ | $32 \pm 15$ |
| RIDE | $2467 \pm 456$ | $1876 \pm 431$ | $1651 \pm 325$ | $92 \pm 31$ | $570 \pm 45$ | $65 \pm 16$ |
| NovelD | $1785 \pm 423$ | $649 \pm 145$ | $975 \pm 360$ | $-134 \pm 27$ | $258 \pm 49$ | $13 \pm 11$ |
| E3B | $1025 \pm 347$ | $1474 \pm 129$ | $1997 \pm 115$ | $66 \pm 27$ | $518 \pm 23$ | $43 \pm 17$ |
| LIBERTY | $\mathbf{2973 \pm 437}$ | $\mathbf{2479 \pm 315}$ | $2766 \pm 487$ | $\underline{292 \pm 68}$ | $\mathbf{681 \pm 73}$ | $\mathbf{73 \pm 21}$ |
| EME | $\underline{2779 \pm 412}$ | $\underline{2369 \pm 314}$ | $\mathbf{2785 \pm 399}$ | $\mathbf{539 \pm 62}$ | $\underline{599 \pm 65}$ | $\mathbf{77 \pm 18}$ |

| Methods | Delay = 20 | | | | | |
|---|---|---|---|---|---|---|
| | HalfCheetah | Hopper | Walker2d | Ant | Humanoid | Swimmer |
| ICM | $1185 \pm 287$ | $1097 \pm 275$ | $995 \pm 201$ | $-175 \pm 23$ | $434 \pm 48$ | $23 \pm 10$ |
| RND | $1595 \pm 415$ | $1925 \pm 401$ | $1379 \pm 193$ | $127 + 12$ | $\underline{519 \pm 25}$ | $29 \pm 12$ |
| RIDE | $2285 \pm 402$ | $1621 \pm 382$ | $\underline{1724 \pm 307}$ | $105 \pm 27$ | $509 \pm 21$ | $59 \pm 13$ |
| NovelD | $945 \pm 355$ | $513 \pm 86$ | $794 \pm 320$ | $-107 \pm 29$ | $315 \pm 55$ | $17 \pm 11$ |
| E3B | $887 \pm 242$ | $1015 \pm 185$ | $743 \pm 95$ | $381 \pm 45$ | $321 \pm 36$ | $20 \pm 11$ |
| LIBERTY | $\mathbf{2619 \pm 354}$ | $\underline{2112 \pm 208}$ | $\mathbf{2345 \pm 414}$ | $\underline{263 \pm 55}$ | $\mathbf{617 \pm 53}$ | $\mathbf{67 \pm 18}$ |
| EME | $\underline{2607 \pm 276}$ | $\mathbf{2467 \pm 298}$ | $1718 \pm 163$ | $\mathbf{352 \pm 35}$ | $499 \pm 62$ | $\underline{59 \pm 15}$ |

| Methods | Delay = 30 | | | | | |
|---|---|---|---|---|---|---|
| | HalfCheetah | Hopper | Walker2d | Ant | Humanoid | Swimmer |
| ICM | $1017 \pm 276$ | $965 \pm 213$ | $798 \pm 199$ | $-198 \pm 25$ | $417 \pm 45$ | $19 \pm 11$ |
| RND | $1483 \pm 393$ | $1773 \pm 391$ | $1038 \pm 191$ | $99 + 13$ | $\underline{501 \pm 27}$ | $24 \pm 11$ |
| RIDE | $\underline{1973 \pm 369}$ | $1405 \pm 315$ | $1345 \pm 305$ | $87 \pm 21$ | $487 \pm 25$ | $\underline{41 \pm 15}$ |
| NovelD | $885 \pm 217$ | $505 \pm 148$ | $664 \pm 129$ | $-205 \pm 43$ | $279 \pm 65$ | $18 \pm 10$ |
| E3B | $997 \pm 159$ | $1246 \pm 101$ | $1007 \pm 142$ | $\underline{221 \pm 25}$ | $366 \pm 26$ | $19 \pm 12$ |
| LIBERTY | $273 \pm 317$ | $\underline{1873 \pm 228}$ | $\mathbf{2077 \pm 398}$ | $\underline{215 \pm 48}$ | $\mathbf{587 \pm 63}$ | $\underline{52 \pm 15}$ |
| EME | $\mathbf{2448 \pm 263}$ | $\mathbf{1999 \pm 288}$ | $\underline{1685 \pm 158}$ | $\mathbf{315 \pm 29}$ | $495 \pm 36$ | $\mathbf{58 \pm 13}$ |

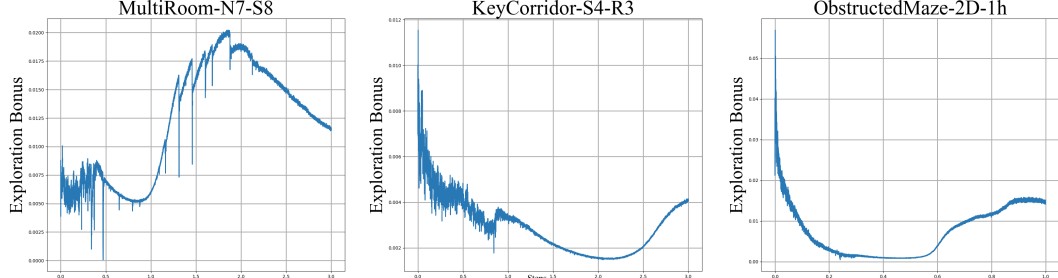

Figure 11: The scale of exploration bonus across all MiniGrid environments

## C.4 Delayed Reward Setting of Mujoco

We present results from the delayed reward setting in the MuJoCo environment [21], following the methodology outlined in [63], where accumulated rewards are delayed every 10, 20, and 30 steps. The experiments assess six tasks: HalfCheetah, Hopper, Walker2d, Ant, Swimmer, and Humanoid. As shown in Table 4, our method achieves the best or second-best performance in 15 out of 18 delayed reward tasks. This indicates that EME can facilitate effective exploration and maintain high performance even under sparse reward conditions. Notably, as rewards become sparser, EME's performance improves and becomes more robust. In contrast, other metric-based exploration bonus methods such as RIDE and NovelD struggle due to their less expressive metrics. Curiosity-driven methods like ICM and RND also become less effective as rewards grow sparser. These results provide further evidence of EME's ability to promote efficient exploration effectively, even in environments characterized by delayed rewards.

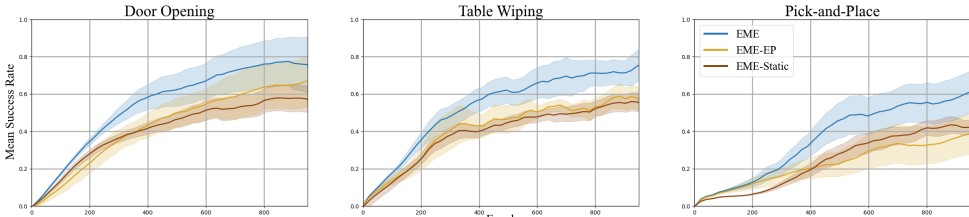

Figure 12: Results of EME, EME with a static scaling factor, and EME with an episodic count from Robosuite. The x-axis represents the number of epochs in training. The y-axis represents the mean success rate(standard deviations in shade).

## C.5 Ablation Study on Scaling Factor

To demonstrate the effectiveness of our proposed scaling factor, we conducted an ablation study focusing on this component. We compared EME with two variants: 'EME-EP,' which incorporates episodic counts, and 'EME-Static,' which uses a static scaling factor. The results from continuous tasks in the Robosuite and MiniGrid environments are illustrated in Figure 12, Figure 9 respectively. Without the diversity-enhanced scaling factor, EME shows a noticeable decline in performance. 'EME-EP' performs better in MiniGrid environments compared to continuous tasks in Robosuite, highlighting the efficacy of episodic state visitation counts in grid-based games but demonstrating limited effectiveness in environments with high-dimensional states. Additionally, we visualize the exploration bonus of all MiniGrid environments in Figure 11.

## D  Implementation Details

### D.1  Environments Settings

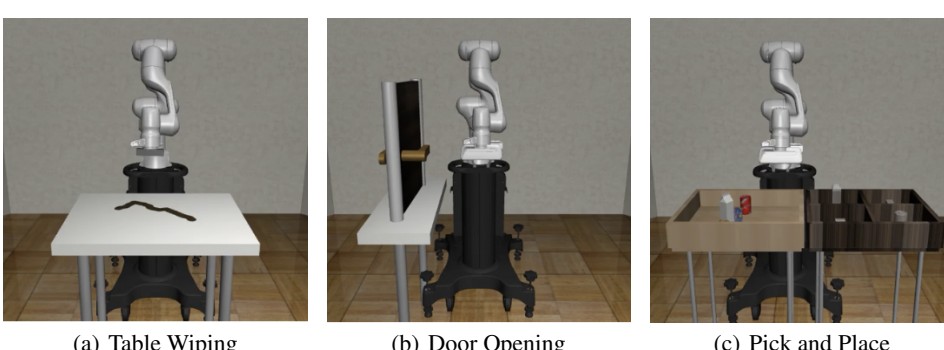

| (a) Table Wiping | (b) Door Opening | (c) Pick and Place |

### D.1.1  Robosuite Manipulation Tasks

We choose the three manipulation tasks in the Robosuite platform [70].

**Table Wiping.** The environment consists of a table with a whiteboard surface and some markings is placed in front of a single robot arm, which has a whiteboard eraser mounted on its hand. The shape of the dirty region is correlated to the position of the cube. When the dirty region is diagonal, the cube is on the right-hand side of the robot arm. The goal of the agent is to learn to wipe the whiteboard surface and clean all of the markings. The whiteboard markings are randomized at the beginning of each episode.

**Door Opening.** The environment consists of a door with a handle mounted in free space in front of a single robot arm. The height of the handle and the position of the door are correlated. When the door is closed to the robot arm, the handle is in a low position. When the door is far from the robot arm,

the handle is in a high position. The goal of the agent is to learn to turn the handle and open the door. The initial state distribution of the door location is randomized at the beginning of each episode.

**Pick and Place.** The environment consists of four objects placed in a bin in front of a single robot arm. There are four containers next to the bin. The goal of the agent is to place each object into its corresponding container. This task also has easier single-object variants. The initial state distribution of object locations is randomized at the beginning of each episode.

### D.1.2 Habitat

The Habitat platform relies on several key abstractions that model the domain of embodied agents and the tasks they can perform in three-dimensional indoor simulation environments.

This platform consists of a virtually embodied agent, such as a robot, equipped with a suite of sensors that can observe the environment and take actions to alter the agent's state or the environment's state. Each sensor is associated with a specific agent and can return observation data from the environment at a specified frequency. The 3D environment includes a scene mesh, objects, agents, and sensors, organized into regions and objects through a hierarchical representation called the Scene, which can be programmatically manipulated. All components of the Scene are present on

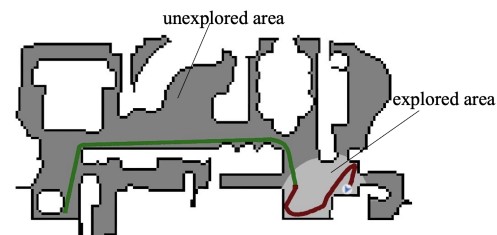

Figure 13: Explored area in Habitat

the SceneGraph. Additionally, a simulator backend instance can update the state of agents and SceneGraphs based on given actions for a set of configured agents and SceneGraphs, and it provides observations for all active sensors possessed by the agents. We use the Matterport3D (MP3D) dataset [15], which consists of high-quality renderings of indoor scenes. To measure exploration coverage, we compute the area revealed by the agent's line of sight using the function provided by the Habitat codebase [32]. As depicted in Figure 13, exploration is measured as the proportion of the environment revealed by the agent's line of sight over the course of the episode.

### D.1.3 MiniGrid Environments

MiniGrid [17] is a set of procedurally-generated grid-worlds. In MiniGrid, the world is a partially observable grid of size $N \times N$. Each tile in the grid contains exactly zero or one object. The possible object types are wall, door, key, ball, box, and goal. Each object in MiniGrid has an associated discrete color, which can be one of red, green, blue, purple, yellow, or grey. By default, walls are always grey and goal squares are always green. Rewards are sparse for all MiniGrid environments. There are seven actions in MiniGrid: turn left, turn right, move forward, pick up an object, drop an object, toggle and done. The agent can use the turn left and turn right action to rotate and face one of 4 possible directions (north, south, east, west).

The move forward action makes the agent move from its current tile onto the tile in the direction it is currently facing, provided there is nothing on that tile, or that the tile contains an open door. The agent can open doors if they are right in front of it by using the toggle action. Observations in MiniGrid are partial and egocentric. By default, the agent sees a square of $7 \times 7$ tiles in the direction it is facing. These include the tile the agent is standing on. The agent cannot see through walls or closed doors. The observations are provided as a tensor of shape $7 \times 7 \times 3$. However, note that these are not RGB images. Each tile is encoded using 3 integer values: one describing the type of object contained in the cell, one describing its color, and a flag indicating whether doors are open or closed. This compact encoding was chosen for space efficiency and to enable faster training. For all

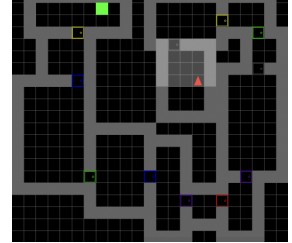

Figure 14: Rendering of the MultiRoomN12S10 in MiniGrid.

tasks, the agent gets an egocentric view of its surroundings, consisting of $3 \times 3$ pixels. A neural network parameterized as a CNN is used to process the visual observation.

**MultiRoom.** The MultiRoomNXSY environment consists of X rooms, with a size at most Y, connected in random orientations. The agent is placed in the first room and must navigate to a green

goal square in the most distant room from the agent. The agent receives an egocentric view of its surroundings, consisting of $3 \times 3$ pixels. The task increases in difficulty with X and Y. Episodes finish with a positive reward when the agent reaches the green goal square. Otherwise, episodes are terminated with zero reward after a maximum of $20 \times N$ steps.

**KeyCorridor.** the agent has to pick up an object which is behind a locked door. The key is hidden in another room, and the agent has to explore the environment to find it. Episodes finish with a positive reward when the agent picks up the ball behind the locked door or after a maximum of 270 steps.

**ObstructedMaze.** The agent has to pick up a box which is placed in a corner of a maze. The doors are locked, the keys are hidden in boxes and the doors are obstructed by balls. Episodes finish with a positive reward when the agent picks up the ball behind the door or after a maximum of 576 steps.

## D.2 Algorithm Details

---

**Algorithm 1** Effective Metric-based Exploration-bonus (EME)

---

1: Initialize parameters of policy $\pi(\theta)$, metric $d_E^\phi$, and reward models $\{g(\eta_1), \ldots, g(\eta_k)\}$
2: **while** *not converged* **do**
3:     **for** $t = 1$ to MAX_STEP_PER_EPISODE **do**
4:         Sample action $a_t \sim \pi_\theta (a_t \mid s_t)$
5:         Step environment $s_{t+1} \sim p (s_{t+1} \mid s_t, a_t)$
6:         Record transition in the buffer: $\mathcal{D}_\tau \leftarrow \mathcal{D}_\tau \cup \{s_t, a_t, s_{t+1}, r_{t+1}\}$
7:         Sample a mini batch with size $B \sim \mathcal{D}$ from the buffer
8:         Compute the variance $\zeta(r_{s_t})$ as scale factor from the ensemble of reward models
9:         Compute bonus $b_{t+1} = d_E(s_t, s_{t+1}) * min\{max\{\zeta(r_{s_t}), 1\}, M\}$
10:        Reshape reward with the bonus $r'_{t+1} = r_{t+1} + b$
11:        Train policy $\pi(\theta)$ via policy gradient
12:        Update the metric $d_E^\phi$: $\mathbb{E}_B[J(\phi)]$    ▷Equation (9)
13:        Update reward models by minimizing the prediction error: $\mathbb{E}_{\mathcal{D}_\tau} \|g(s_t, a_t; \eta) - r_{t+1}\|^2$
14:     **end for**
15: **end while**

---

## D.3 Computation Details

All of our experiments are conducted on 4 GPUs with 16 CPU threads, which include AMD Ryzen 9 CPU@1.20GHz (16 core) CPU, NVIDIA GeForce GTX 3080Ti GPUs, and 64GB memory.

## D.4 RL Hyperparameters

Table 5: Hyper-parameters in Robosuite experiments

| Parameters | Environment | | |
| --- | --- | --- | --- |
| | Pick and Place | Door Opening | Table Wiping |
| Horizon steps | 300 | 500 | 500 |
| Control frequency (Hz) | 20 | 20 | 20 |
| State dimension | 92 | 22 | 30 |
| Action dimension | 7 | 7 | 6 |
| Controller type | OSC position | Joint velocity | Joint velocity |
| Maximum reward scaling | 10 | 10 | 20 |
| Actor learning rate | $3 \times 10^{-4}$ | - | - |
| Critic learning rate | $1 \times 10^{-3}$ | - | - |
| Batch size | 128 | - | - |
| Discount factor | 0.99 | - | - |
| Soft update weight | 0.005 | - | - |
| Alpha learning rate | $3 \times 10^{-4}$ | - | - |
| Hidden layers | [256, 256, 256] | - | - |
| Returns estimation step | 4 | - | - |
| Buffer size | $1 \times 10^{6}$ | - | - |
| Steps per update | 10 | - | - |

Table 6: Common Hyperparameters for MiniGrid

| Parameter | Value |
| --- | --- |
| Learning Rate | 0.0001 |
| RMSProp momentum | 0 |
| RMSProp $\epsilon$ | $10^{-3}$ |
| Unroll Length | 100 |
| Number of buffers | 80 |
| Number of learner threads | 4 |
| Number of actor threads | 256 |
| Max gradient norm | 40 |
| Entropy Cost | 0.0005 |
| Baseline Cost | 0.5 |
| Discounting Factor | 0.99 |
| Maximum reward scaling | 5 |

Table 7: Common Hyperparameters for Habitat

| Parameter | Value |
| --- | --- |
| Clipping | 0.2 |
| PPO epochs | 2 |
| Number of minibatches | 16 |
| Value loss coefficient | 0.5 |
| Entropy coefficient | 5e-5 |
| Learning rate | 7e-5 |
| $\epsilon$ | $10^{-5}$ |
| Max gradient norm | 0.8 |
| Rollout steps | 128 |
| Use GAE | True |
| $\gamma$ | 0.99 |
| $\tau$ | 0.95 |
| Use linear clip decay | True |
| Maximum reward scaling | 30 |
| Hidden size | 512 |

| Algorithm | Hyperparameters |
|---|---|
| Base Learner (PPO) | Maximum reward scaling: 10
threshold of probability ratio clipping $(\epsilon)$ : 0.5
update timesteps: 2e10
number of epoches per update: 50
number of minibatches: 16
batch size: 2048
GAE parameter $(\lambda)$ : 0.95
optimizer: Adam
learning rate: $5 \times 10^{-3}$
policy gradient clip norm: 0.9
discount rate $(\gamma)$ : 0.99 |
| E3B | ridge regularizer $\lambda$: 0.1
tanh activation
entropy cost: 0.005
intrinsic reward coefficient: 1.0
gradient clip norm: 1.0 |
| ICM | forward model loss coefficient: 0.2
inverse model loss coefficient: 0.08
entropy cost: $10^{-4}$
intrinsic reward coefficient $\beta$: 0.005 |
| RND | proportion of experience used for training predictor: 0.25
predictor Model updates per PPO epoch: 6
entropy cost: 0.005
intrinsic reward coefficient $\beta$: 0.1
forward model loss coefficient: 0.5 |
| RIDE | inverse model loss coefficient: 0.8
entropy cost: 0.0005
intrinsic reward coefficient $\beta$: 0.1 |
| NovelD | episodic memory capacity: 5000
action prediction network filter sizes: (3,3)
entropy cost: 0.005
RND clipping factor $L$: 5
intrinsic reward coefficient $\beta$: 1.0 |
| LIBERTY | intrinsic reward coefficient $\beta$: 0.05
entropy cost: 0.005
inverse model loss coefficient: 0.5
forward model loss coefficient: 0.5 |

Table 8: The hyperparameters of the tested benchmark algorithms in the Atari experiment

### D.5 Codebases Used

Our codebase was built atop the following codebases:

- The official NovelD codebase: https://github.com/tianjunz/NovelD (Creative Commons Attribution-NonCommercial 4.0 license) for NovelID, RND, RIDE and count-based baselines (this codebase is built atop the official RIDE codebase below)
- The official RIDE codebase: https://github.com/facebookresearch/impact-driven-exploration (Creative Commons Attribution-NonCommercial 4.0 license)
- The official LIBERTY codebase: https://github.com/Mingle0228/liberty (MIT License)
- The official E3B codebase: https://github.com/facebookresearch/e3b(Creative Commons Attribution-NonCommercial 4.0 license)

## E  Limitation

While our method has shown success in several challenging exploration tasks, it does not consistently achieve top performance in environments with less sparse rewards, as observed in the MuJoCo continuous control benchmarks. Additionally, our method has not been tested in reinforcement learning domains characterized by large action spaces. Addressing these limitations and developing a more generalized solution for diverse RL environments remains a goal for future work.

## F  Related Work

**Exploration in RL.** Exploration remains a long-standing problem in RL. Common approaches include $\epsilon$-greedy [59], count-based exploration [7, 46, 61, 41, 40, 57], ensemble-based exploration [45, 39, 48] and curiosity-based exploration [53, 55, 56, 10]. Several exploration strategies use a dynamics model to provide intrinsic rewards [47, 11, 33, 48, 36, 2]. Latent variable dynamics have also been studied for exploration [4, 9, 62]. Maximum entropy in the state representation has also been used for exploration [54, 66]. Other intrinsic motivation methods have recently been developed for exploration in context MDPs [67, 28, 51, 69, 32], which automatically generate curricula over variations of the MDP to encourage efficient learning, effectively performing a form of curiosity-driven exploration in the context space, including goal-conditioned [23, 50, 12, 18, 22] and goal-free variants [58, 49, 34, 19]. Reward shaping refers to modifying the original reward function with a shaping reward function which incorporates domain knowledge. Considering the most general form, namely the additive form, of reward shaping. The first approach to guarantee policy invariance is potential-based reward shaping (PBRS) [44], which defines the shaping reward function as the difference between values assessed through the potential function based on prior knowledge. There are lots of variants of PBRS [20, 38, 31, 3, 64, 30, 29, 63] which have shifted their focus to different areas within the field of reinforcement learning.

**Metric-based Exploration Bonus.** Metric space is widely used in encoding state representations [68, 14, 13, 16, 35]. Metric-based exploration bonus is based on the evaluation metric that quantifies the degree of difference or distance between two projected states for the measure of novelty. RIDE [51] evaluates the novelty as between two successive state representations under the $L_2$ norm. NovelD [69] uses the disparity in RND bonuses between adjacent states under the $L_1$ distance as the exploration bonus. LIBERTY [63] uses the difference between potential functions of adjacent states under the bisimulation metric space. Note that we propose an effective metric that measures the distribution distance between dynamics models by computing the distance between sampled subsequent states. Additionally, we integrate the Kullback–Leibler (KL) divergence between policy distributions to more robustly model the "**behavioral similarity**" between states, which is more effective and scalable across diverse environments with different observations, especially in addressing the critical Noisy-TV problem during exploration.

