# OpenReview forum: "Rethinking Exploration in Reinforcement Learning with Effective Metric-Based Exploration Bonus"
_NeurIPS.cc/2024/Conference — NeurIPS 2024 spotlight_

### Official Review · Reviewer_DhVP · 2024-07-08

**Soundness:** 3
**Presentation:** 3
**Contribution:** 3
**Rating:** 7
**Confidence:** 5

**Summary:**

The authors propose a robust metric for state discrepancy evaluation backed by comprehensive theoretical analysis. Previous exploration algorithms measure state discrepancy strictly on $L_p$ norms which is limiting since for environments where state differences are minimal in the observations the intrinsic reward is not useful. The authors propose an exploration bonus which is centred aroudn the variance of the predicted reward for a ensemble of reward models. A scaling factor for the exploration bonus is also introduced using the ensemble of reward models, where the variance in the prediction of the ground-truth reward is taken as the scaling factor. This scaling factor is higher in novel spaces since the prediction variance increases for states that have not been explored sufficiently.. The authors conduct extensive experiments on 2D and 3D environments shows superior performance to other known exploration methods.

**Strengths:**

- The paper is well written and structured, although a rough overview of the components of the exploration bonus at the beginning would have further helped my understanding.
- The approach to consider the metric in which state discrepancy is measured is an interesting idea and seems fitting and quite novel.
- The intrinsic reward is based on the discrepancy of an ensemble of reward models instead of the observations which is usually done, which I find an interesting idea.
- The code is available and of high quality which should allow for reproducibility of the results
- The authors motivate their method very well, highlighting why previous methods perform worse.

**Weaknesses:**

- To my knowledge other methods that try to aggregate states in high dimensions are not considered. Wouldn’t this mitigate or solve the issue with unique counts in high-dimensional spaces?
- The maximum reward scaling $M$ seems to be an important parameter, however I do not see an ablation for it. The ablations in the appendix ablate the nature of the scaling factor, but not $M$
- To my knowledge there is no talk about limitations of the method. Could you provide insight on where possible limitations to your method lie?

**Questions:**

- In practice, how do you train the ensemble of reward models in the reward free setting? Is there any relation to successor representations? What do you do in settings where the reward is not a good indicator for the environment structure? I'm a bit surprised that the method works well given there is no reward to learn the ensemble models. Does the KL-divergence of the policies take the overhand here?

**Limitations:**

The authors have discussed limitations of previous methods but have not to my knowledge discussed the limitations of their own method.

---

> ### Author Rebuttal · Authors · 2024-08-06
>
> We gratefully thank the reviewer for recognizing our contributions, writing and novelty. The additional ablation experiments have been included in the PDF file and all the references mentioned in the response can be found in the reference list in the global comment. We provide our response below:
>
> *W1: To my knowledge other methods that try to aggregate states in high dimensions are not considered. Wouldn’t this mitigate or solve the issue with unique counts in high-dimensional spaces?*
>
> **Response**: Thank you for the insightful question. Indeed, one alternative to address the issue of unique counts in high-dimensional spaces is to design a function that extracts relevant features from each state and feeds them into the count-based episodic bonus. Specifically, instead of defining a bonus using $N_e(s_t)$, we could define the bonus using $N_e(\phi(s_t))$, where $\phi$ is a hand-designed feature extractor. In our experiments, following the RIDE implementation in [G], in the Habitat environment, the RIDE and NovelD implementations use $\phi(s_t) = (x_t, y_t)$, where $(x_t, y_t)$ is the spatial location of the agent at time t, so the unique state count issue is solved. However, despite this, the performance of RIDE and NovelD still lags behind in Habitat and the state counts is still very sparse. Additionally, the aggregation of states using a feature extractor relies heavily on task-specific knowledge, which significantly restricts scalability. Another approach involves using pseudo-counts [H], which estimate counts in high-dimensional state spaces using a density model. This method is adopted by RIDE and NovelD in our MiniGrid experiments and the performance still falls behind ours. Consequently, even with pseudo-counts, the training efficacy and accuracy of the estimated counts remain challenging issues. Overall, even if we solve the unique count issue, the performance of count-based bonuses still struggles in high-dimensional environments due to the sparsity and limited accuracy of counts.
>
> *W2: The maximum reward scaling seems to be an important parameter, however I do not see an ablation for it. The ablations in the appendix ablate the nature of the scaling factor, but not M*
>
> **Response**: Thank you for the valuable suggestions. Additional experiments can be found in the uploaded pdf. We provide an ablation study on the value of M in Figure 2 for continuous control tasks in Robosuite. M sets an upper limit on the bonus. We set M=10 as the default setting. As we can see, if M=1, the scaling factor is fixed to 1, resulting in a significant performance decline. Higher M encourages more extensive exploration. The performance with M=5 slightly lags behind the default setting. The performance with M=20 and M=40 is comparable and almost the same, indicating that the performance stabilizes as M increases. Practically, the value of M can be adjusted depending on the specific task and environment to determine the intensity of exploration. We are glad to include the detailed ablation experiment in the revision.
>
> *W3:To my knowledge there is no talk about limitations of the method. Could you provide insight on where possible limitations to your method lie?*
>
> **Response**: Thank you for the observation. We want clarify that we have provided the cross-reference to limitations in the paper checklist (see Appendix E). This discussion was acknowledged by reviewers 7tZv, TUTk, and tp7G. To summarize briefly, our method does not consistently achieve top performance in environments with less sparse rewards with untested performance in RL domains with very large action spaces.
>
> *Q1: In practice, how do you train the ensemble of reward models in the reward free setting? Is there any relation to successor representations? What do you do in settings where the reward is not a good indicator for the environment structure? I'm a bit surprised that the method works well given there is no reward to learn the ensemble models. Does the KL-divergence of the policies take the overhand here?*
>
> **Response**:Thanks for the insightful question! First, we want to emphasize that in the reward-free setting of Habitat, since there is no reward, we compare the exploration capability of these methods by evaluating the portion of explored space. Therefore, the **exploration bonus** takes the lead here. The performance can be analyzed from two perspectives:
>
> - First, regarding the scaling factor, in the reward-free setting for training the ensemble models, each model’s parameters are initialized differently, and we train each model on a subset of data randomly sampled with replacement. Without the external reward, most labels of the training samples are zero during the training process, leading to larger variance between predictions during the training phase. The increased variance requires more time or a life-long learning approach for the ensemble models to converge, allowing for a larger scaling factor in such scenarios to promote more extensive exploration.
>
> - Second, from the point of metric, referring to EME metric definition in Equation (4), when the reward is zero, the KL-divergence of the policies indeed plays a significant role. It encourages agents to take more diverse actions, compensating for the lack of a reward signal.
>
> Overall, based on our bonus Equation (10), the EME bonus can promote more extensive exploration compared to other methods when the reward signal is sparse or not a good indicator of the environment structure. The effectiveness is also evidenced in Figure 8, which demonstrates the performance of EME in reward-free experiments. Thus, while the KL-divergence does take a more prominent role in these settings, it effectively drives the exploration process combined with diversity-enhanced scaling factor, ensuring the method remains robust and efficient. We are glad to include the discussion in the main paper to further improve the clarity.

---

> > ### Comment · Reviewer_DhVP · 2024-08-11
> >
> > Thank you so much for your response and added work to the ablations!
> >
> > I feel you have addressed all my concerns and also have a better understanding as to why the scaling factor and EME metric work in a reward free setting.
> >
> > I think this is a good paper with good experiments.

---

> > > ### Author Response · Authors · 2024-08-11
> > >
> > > Thank you once again for recognizing our work and investing your valuable time in providing feedback on the paper, your valuable suggestions will definitely improve the quality of our work.

---

### Official Review · Reviewer_tp7G · 2024-07-15

**Soundness:** 3
**Presentation:** 2
**Contribution:** 3
**Rating:** 7
**Confidence:** 4

**Summary:**

This paper proposes a new exploration algorithm in reinforcement learning (RL), particularly in environments with sparse rewards. The authors critique existing exploration bonus methods using state discrepancy, highlighting their limitations in scalability and theoretical guarantees. They propose a novel method called Effective Metric-based Exploration-bonus (EME), which features a robust metric for evaluating state discrepancy, grounded in theoretical assurances. EME avoids approximation gaps found in previous bisimulation metric-based approaches and eliminates the reliance on episodic count scaling factors.  Furthermore, they introduce a diversity-enhanced scaling factor, dynamically adjusted by the variance of predictions from an ensemble of reward models, to improve exploration in tasks with subtle state differences. Experiments on Robosuite, Atari games, MiniGrid, and Habitat environments demonstrate the effectiveness and scalability of EME.

**Strengths:**

*Originality*: The EME metric presents a novel approach for measuring state discrepancies. The paper adequately cites related work and differentiates its contributions from existing methods.

*Quality*: The paper provides theoretical analysis to support the claims regarding the limitations of current methods and the guarantees offered by the EME metric. The experimental evaluation is conducted across different domains, supporting the effectiveness of the proposed method.

*Clarity*: The paper is generally well-written and organized. The motivation and background information are clearly presented, and the proposed method is explained in detail.

*Significance*: The results shows good performance improvement over existing algorithms and show that EME could be become the default exploration algorithm for most problems.

**Weaknesses:**

While the presented results look impressive a more thorough empirical evaluation would be more helpful to assess the performance of EME:

- On Atari the paper only evaluates on hard exploration games though it has been shown that evaluating on more Atari games is usually necessary to properly assess the performance of an algorithm. [1]
- The paper does not include a ablation, because of that it is difficult to know if the improvement comes from the better exploration bonus and using multiple reward predictions

[1] On Bonus-Based Exploration Methods in the Arcade Learning Environment, Ali Taiga et al. ICLR 2020

**Questions:**

- The explanation in lines 99-102 regarding the breaking of theoretical assurances is unclear. Can the authors elaborate on what specifically breaks down and how EME overcomes this issue?

- On L130, the authors mention that existing methods might already be performing a similar function.  Can the authors clarify this statement and clearly differentiate EME from such methods?

- Why is the Habitat environment considered a more realistic scenario given the variety of task one could attempt to solve with reinforcement learning?

- The paper lacks citations for using an ensemble of models to drive exploration (L261), see for example [1, 2]

- Is the feature encoder jointly learned with the EME metric? I couldn't find the information, though results in the appendix show that even a random feature encoder is providing good results

- How does the number of reward models in the ensemble affect EME's performance and computational cost? Is there an optimal number, or are there diminishing returns as the ensemble size increases?


[1] Deep Exploration via Bootstrapped DQN, Osband et al.

[2] The Curse of Diversity in Ensemble-Based Exploration, Lin et al. ICLR 2024

**Limitations:**

The authors acknowledged limitations regarding the performance in environments with less sparse rewards and untested performance in RL domains with large action spaces.
An ablation study of the different component of the algorithm as well as a more thorough evaluation on diverse environments would be helpful to properly evaluation the contribution presented in this paper.
The potential negative societal impact is also addressed in the broader impact statement

---

> ### Author Rebuttal · Authors · 2024-08-06
>
> We would like to thanks to the reviewer for their insightful feedback. Additional exps have been included in the uploaded pdf, for the references mentioned in the response, please find the reference list in the global comment and we provide our response below:
>
> *W1:(1)evaluating on more Atari games..(2)does not include a ablation..*
>
> **Response**: Thank you for the valuable suggestions. The results of more Atari games is in Fig 1 of PDF. We selected 10 widely used Atari games based on [C,D]. EME still achieves top performance and achieves the best performance in 8 out of 10 games. For the ablations, we have already provided results of scaling factor in Appendix C.4, as shown in Fig 8, 10, and 11. For metric comparison, we also include the performance of other metric-based methods with our scaling factor to isolate the impact of our scaling factor, denoted as LIBERTY with DSF and RIDE with DSF, and the results are shown in Fig 1 of the PDF. EME still achieves the best performance, indicating the superiority of our metric. Additionally, we provide further ablation studies on the ensemble size, maximum reward scaling M, and different RL algorithms in Fig 2, 3, and 4 of the PDF. We will include more detailed ablation results of other envs in the revision
>
> *Q1:..line99-102 is unclear..how EME overcomes the issue?*
>
> **Response**: We are afraid that there is some mistake in the line number 99-102. We assume that you may be unclear about the statement "breaking theoretical integrity of bisimulation metric", and our response are provided as follows. First we have clarified the statement in Propostion 1. The relaxation of Wasserstein distance $W_1$ to $W_2$ will break the theoretical guarantee of bisimulation metric when the transition dynamic model or the policy is stochastic, e.g. the fixed-point guarantee is broken under the circumstance, the metric may be not able to converge in the training process. EME overcomes the issue by eliminating the need of calculating $W_1$ distance and preserves the guaranteed fixed-point and value difference bound theoretical property, as stated in Theorem 1 and 2. The detailed proof can be found in Appendix B.
>
> *Q2:..L130..differentiate EME from such methods?*
>
> **Response**: Again we are afraid that there is some mistake in the line number 130. We assume that you are referring to line 91-92 that state discrepancy-based exploration bonus methods can be defined as Equation (1). These methods' bonuses can generally be defined as the product of a measure of state discrepancy and a scaling factor. What differentiates EME from such methods is twofold. First, EME uses a novel metric that more effectively and robustly captures state similarity backed by thm 1 and 2. Second, we propose a dynamic diversity-enhanced scaling factor, whereas the scaling factors of previous methods are either not scalable or are hand-crafted parameters. This combination allows EME to adaptively balance exploration and exploitation, leading to improved performance. For a detailed comparison, please refer to Table 1.
>
> *Q3:Why is Habitat a more realistic scenario?*
>
> *Q4:..lacks citations..[1,2]*
>
> **Response**: Thanks for pointing out the citations! we will include them in the revision. We had included a detail introduction of Habitat in Appendix D.1.3. Habitat is a platform designed for embodied AI research, where agents navigate and act within photorealistic simulations of real indoor environments. It is recognized as a more realistic indoor scenario in the RL research community, as noted in [E,F].
>
> *Q5: Is the feature encoder jointly learned.. random encoder shows good results*
>
> **Response**: We are sorry for the confusion. The feature encoder is pre-trained to study how EME performs under different representations. When using bisimulation and inverse dynamic encoders, the exploration bonus is restricted because their representations are very compact, making state differences subtle and thus harming exploration efficacy. In contrast, as also oberseverd in experiments of [I], random embedded states introduce randomness into the calculation of the exploration bonus, sometimes resulting in higher bonuses in sparse reward environments. This can occasionally provide good results but also introduces higher variance, as shown in Fig 6 and 7.
>
> *Q6:How does number of reward models affect performance..*
>
> **Response**: Thank you for the valuable question. We carry the ablation study on ensemble size in Figure 3 in pdf. From the study, we observe that as the ensemble size (ES) increases, the performance of ES = 6 and ES = 9 surpasses that of ES = 3. The performance of ES = 6 and ES = 9 is comparable, with no significant difference. However, when the size is further increased to ES = 12, there is a decline in performance, particularly in the Table Wiping and Pick and Place tasks. The performance change can be analyzed from two perspectives. First, the number of reward models is related to the accuracy of the reward variance prediction used to calculate the loss function. As the ensemble size increases, the overall prediction error decreases because the models can average out individual errors more effectively. This leads to a more accurate approximation of the variance $\zeta(s_t)$. Second, with a larger ensemble size, the variance of reward predictions decreases, resulting in a lower scale of the exploration bonus. Thus, there is a balance or trade-off between estimation accuracy and exploration, explaining why the performance of ES = 12 lags behind ES = 9. The optimal number of ensemble models may vary depending on the specific tasks and environments. Regarding computational cost, it increases with the ensemble size. Therefore, we set ES = 6 as our default setting, where the performance is nearly the same as ES = 9 but with lower computational overhead. It is also noteworthy that even with an ensemble size of 3, EME still outperforms the best baseline methods, further demonstrating the robustness of EME.

---

> > ### Comment · Reviewer_tp7G · 2024-08-12
> >
> > Thank you for your response and properly addressing my comments. I have increased my score to 7.
> > If you have enough time/compute I would encourage you to add the remaining atari games to your results.

---

> > > ### Author Response · Authors · 2024-08-13
> > >
> > > Thank you once again for recognizing our work and investing your valuable time in providing feedback on the paper, we will work on more atari games, and your valuable suggestions will definitely improve the quality of our work.

---

### Official Review · Reviewer_TUTk · 2024-07-16

**Soundness:** 3
**Presentation:** 3
**Contribution:** 3
**Rating:** 7
**Confidence:** 4

**Summary:**

This paper introduces the Effective Metric-based Exploration-bonus (EME), a novel approach for enhancing exploration in reinforcement learning (RL) tasks. The paper identifies key limitations in existing metric-based exploration methods, such as their reliance on count-based scaling factors and approximation gaps in bisimulation metric learning. To address these issues, the paper proposes a more robust metric for evaluating state discrepancy that provides theoretical guarantees on the value difference bound. Additionally, they introduce a diversity-enhanced scaling factor based on the variance of predictions from an ensemble of reward models to improve exploration effectiveness, especially in hard exploration scenarios. The authors conduct extensive experiments across various challenging environments, including continuous control tasks, discrete-action games, and realistic indoor scenarios, demonstrating the superior performance and scalability of their method compared to prior approaches.

**Strengths:**

1. This paper includes a nicely-written and comprehensive analysis of the limitations in existing metric-based exploration methods; this really helps setting up the motivation for EME.

2. Extensive experimental evaluation across a diverse set of environments, including continuous control, discrete-action games, and realistic indoor scenarios, demonstrating the effectiveness and scalability of EME.

3. Introduction of a theoretically grounded metric for evaluating state discrepancy, backed by rigorous proofs and guarantees on the value difference bound.

4. The reward-free exploration experiment is very nice because in many real-world scenarios, reward function may be hard to write down.

Overall, this is a solid paper with good methodological contributions and convincing experiment results.

**Weaknesses:**

1. The performance of EME with feature encoder is mixed; this may hamper the scalability of EME to realistic scenarios, where pre-trained feature encoder is commonly used.

2. Ablations are limited. It would be good to know how sensitive EME is to the choice of RL algorithm, the value of M, and other hyperparameters. Only scaling factor is studied right now.

3. On the Robosuite environments, EME do not seem much better than the baselines. How do we characterize when we expect EME to do much better than existing methods?

**Questions:**

My questions are stated above.

**Limitations:**

Yes, the paper has adequately addressed the limitations.

---

> ### Author Rebuttal · Authors · 2024-08-06
>
> We gratefully thank the reviewer for recognizing our contributions! The additional ablation experiments have been included in the uploaded pdf, we provide our response below:
>
> *W1:The performance of EME with feature encoder is mixed; this may hamper the scalability of EME to realistic scenarios, where pre-trained feature encoder is commonly used*
>
> **Response**: Thanks for the valuable question. Yes, we agree that the performance of EME can be influenced by the choice of pre-trained feature encoders, which is also a common issue for existing state difference-based exploration approaches like RIDE where the bonus heavily relies on the design of encoders. However, as demonstrated in Figures 6 and 7, the impact on EME's performance is not significant and can sometimes be positive. Even when different pre-trained encoders are used, EME consistently outperforms other baseline methods, which suggests that despite the inherent variability introduced by different feature encoders, the performance of EME remains at a relatively high level, providing stronger results compared to other methods.
>
> *W2:Ablations are limited. It would be good to know how sensitive EME is to the choice of RL algorithm, the value of M, and other hyperparameters. Only scaling factor is studied right now*
>
> **Response**: Thank you for the valuable suggestions. Additional experiments can be found in the uploaded pdf. We conducted an ablation study on the value of M, shown in Figure 2, for continuous control tasks in Robosuite. M sets an upper limit on the bonus. We set M=10 as the default setting. As we can see, if M=1, the scaling factor is fixed to 1, resulting in a significant performance decline. Higher M encourages more extensive exploration. The performance with M=5 slightly lags behind the default setting. The performance with M=20 and M=40 is comparable and almost the same, indicating that the performance stabilizes as M increases. We also provide the performance of EME with different RL algorithms. We chose widely used benchmark algorithms for continuous control tasks: PPO, A2C, SAC, and DDPG. As shown in Figure 4, EME acts as a plug-in algorithm and demonstrates considerable improvement across all baseline methods. The performance of EME with different RL algorithms is almost at the same level, indicating EME's wide adaptability. Additionally, we conducted an ablation study on ensemble size, presented in Figure 3, and included the performance of other metric-based methods integrated with our proposed scaling factor in Figure 1. These studies show that EME is robust to different parameter choices and consistently demonstrates superior performance compared to other baselines. Due to limited time and space, we will include the detailed ablation experiment of other environments in the revision.
>
> *W3:On the Robosuite environments, EME do not seem much better than the baselines. How do we characterize when we expect EME to do much better than existing methods?*
>
> **Response**: Thanks for the insightful question. We have discussed the point of EME's performance lead in the limitations section in Appendix E. Specifically, we expect EME to outperform existing methods more significantly in environments with sparse rewards and challenging exploration tasks requiring extensive exploration and diverse actions. Examples include navigation tasks in Habitat, hard exploration tasks in Atari and the gym-maze environment. In these settings, where external rewards from the environment are very sparse or zero during the learning process, the advantage of EME's bonus becomes more pronounced compared to other bonuses due to the superiority of our proposed metric and the diversity-enhanced scaling factor.

---

### Official Review · Reviewer_7tZv · 2024-07-21

**Soundness:** 2
**Presentation:** 2
**Contribution:** 2
**Rating:** 6
**Confidence:** 1

**Summary:**

The authors introduce the Effective Metric-based Exploration-bonus (EME) to address the limitations of existing state discrepancy methods by proposing a robust metric for evaluating state differences and a diversity-enhanced scaling factor for exploration bonuses. Extensive experiments demonstrate that EME outperforms other methods in various environments, including Atari games, Minigrid, Robosuite, and Habitat, highlighting its scalability and applicability to diverse scenarios.

**Strengths:**

Originality:

EME has not been proposed before. It appears to be effective.

Quality:

The paper is supported by thorough theoretical underpinnings and extensive experimental validation across many environments.

Clarity:

The paper is well-written and clearly structured.

Significance:

Exploration is a core challenge in RL. This paper makes a significant contribution to it.

**Weaknesses:**

The ultimate method seems incredibly complex, making it difficult for future practitioners to reproduce. Emphasizing Algorithm 1 (potentially moving it from the Appendix into the main paper) could help with this.

The paper does not provide a thorough discussion of the computational efficiency and potential overheads introduced by the EME method. It would be useful to provide a section discussing this.

The paper introduces several novel components, such as the diversity-enhanced scaling factor, but lacks detailed ablation studies that isolate the contributions of these individual components.

The authors repeatedly refer to "a real-life indoor environment". While somewhat true (as it is later revealed to be the Habitat environment), it is rather misleading, as the experiments are still entirely done in the computer. (ie there is no real-world robot involved).

**Questions:**

1. Can you provide more details on the computational complexity and runtime performance of the EME method compared to existing approaches? Specifically, how does EME scale with the size of the state space and the complexity of the environment?

2. Could you provide detailed ablation studies to isolate the impact of the diversity-enhanced scaling factor and other key components of EME?

3. Some parts of the theoretical analysis are dense. Could you provide more intuitive explanations or examples to help bridge the gap between the theory and its practical implications?

**Limitations:**

Yes

---

> ### Author Rebuttal · Authors · 2024-08-06
>
> We appreciate the reviewer for valuable suggestions, the requested exp have been included in the uploaded pdf, for all the references mentioned in the response, please find the reference list in the global comment, and we provide our responseas follows.
>
> *W1:The ultimate method seems incredibly complex..moving Algorithm 1 to main paper could help..*
>
> *Q1:..more details on the computational complexity and runtime performance..how does EME scale with size of the state space..*
>
> **Response**: Thank you for your valuable suggestions. We would like to move Algorithm 1 to the main text in the revision. Regarding the runtime performance, we direct you to Fig 2 and 3 in the paper, which demonstrate that EME achieves higher returns in the same timesteps and faster convergence compared to other baselines, indicating reduced running time during training. In terms of computational complexity comparison, RIDE utilize calculating state difference in representation space (see Table 1), which is learned using both a forward and an inverse dynamics model. While for the loss function of LIBERTY, please refer to Eq (3), we can see that the calculation of LIBERTY requires the approximation of Wasserstein distance of transition distributions and the difference of inverse dynamic output, which requires additional training of transition model and inverse dynamic model. In contrast, as shown in Equation (9), EME simplifies the process by only training additional reward models to calculate the bonus. So our method is more computationally friendly and we require less training procedures comparing to RIDE and LIBERTY.
>
> The performance of EME is scalable because EME provides a more robust and effective metric to evaluate state difference within different environments, and we provide analysis in line 200-212 and 218-226. The scalabity of EME is also evidenced in the experiments where EME outperforms other methods in different environments with different state size. Specifically, when EME scales with environments with different state size or complexity, we only need to adjust the size of network accordingly to fit in the state size for calculation of bonus. Because the objective of metric learning only considers the sum of reward differences, the distance between sampled subsequent states, and the KL divergence between policy distributions, which means that the size of states has minimal impact on the metric learning.
>
> *Q2:Could you provide ablation studies to isolate the components of EME?*
>
> **Response**: Yes, we had already provided the ablation study on scaling factor in Appendix C.4 (line 667-676), where we compared EME with two variants: EME-EP, which incorporates episodic counts, and EME-Static, which uses a static scaling factor. The experimental results are shown in Figure 8, 10 and 11. For metric comparison, we also include the performance of other metric-based methods with diversity-enhanced scaling factor to isolate the impact of our scaling factor, the result is shown in Figure 1 in uploaded pdf, where EME beats other metric-based methods equipped with diversity-enhanced scaling factor, indicating the superiority of our metric comparing to bisimulation metric and $L_p$ norms. Also we provide additional ablation study on the ensemble size, max reward scaling M in Figure 2, 3 and 4 of uploaded pdf. Due to limited space and time, we will include the ablation results with other environments in the revised version.
>
> *Q3:..Could you provide more intuitive explanations..on the theory and its practical implications?*
>
> **Response**: Thank you for your insightful question. Proposition 1 and 2 illustrate the impact of the approximation gap on the bisimulation metric-based method. Fundamentally, the distance derived from the inverse dynamic bisimulation metric could break the fixed-point guarantee, resulting in a less stringent bound on the value difference. This suggests that the metric may not be able to converge and not be closely relate to the value function. Consequently, the learned metric could diverge and fails to precisely reflect state similarity within the bisimulation metric space, leading to a decrease in performance.
>
> Theorem 1 provides a convergence guarantee, stating that our learned metric will converge to a fixed point during training, which ensures that the metric becomes stable and reliable as training progresses. Theorem 2 explains that the EME distance between states serves as an upper bound on the value difference between those states. Practically, this means the agent is encouraged to visit states with both higher state differences and higher value differences. Visiting states with higher value differences enhances the value diversity between collected transition samples, leading to a more diverse policy during training. Furthermore, an increase in bonus reward leads to larger value differences and thus larger TD errors (see lines 218-226), incentivizing the agent to prioritize transitions with large TD errors, so the training efficiency will be improved.
>
> *W2:..a real-life indoor environment..is rather misleading..*
>
> **Response**:Thank you for pointing out the potential for misunderstanding. The detailed description of Habitat is included in Appendix D.1.3. To clarify, we use the Habitat platform [A], which is designed for embodied AI research and provides an interface for agents to navigate and act within photorealistic simulations of real indoor environments. Specifically, we utilize the MP3D dataset[B], which includes high-quality renditions of 1,000 different indoor spaces. This platform offers a practical application for RL in embodied AI, providing a highly realistic and diverse set of environments for testing and development. So we call it a real-life indoor environment. To avoid confusion, we will change the term to "a photorealistic simulator of indoor environments" in our revised manuscript.

---

### Author Rebuttal · Authors · 2024-08-06

We thank the reviewers for their insightful and valuable feedback. Overall, reviwers appreciate the novelty, clarity and contribution of our work, we summarize the major concerns and provide new experimental results with discussions below:

- ### Additional ablation studies and other experiments

According to reviewer 7tZv, detailed ablation studies are required to isolate the impact of the diversity-enhanced scaling factor and other key components of EME. We clarify that we had already included ablation study on diversity-enhanced scling factor in Appendix C.4. Reviewer TUTk and DhVP suggest ablation studies on the maximum reward scaling parameter M. Reviewer tp7G suggests that our method should be tested on more Atari games and propose questions on the impact of ensemble size. To address the concern, we have included additional ablation studies on parameter M, ensemble size, and metric impact comparison to isolate the influence of key components of EME. Additionally, following [C,D], we have chosen 10 other representative Atari games to test our method's performance compared to other baselines. The results, included in the uploaded PDF, show that EME's performance still leads in the majority of games. The analysis of our method's performance and the ablation experiments are provided in the individual responses.

- ### Clarification on theorems and Habitat environment

Reviewer 7tZv suggests to provide more intuitive explanations or examples on the theoretical results, and reviewer tp7G is unclear about how other method break theoretical assurance and how EME overcome the issue. And reviewer 7tZv and tp7G have some questions about why Habitat is called a more realistic environment. We have clarified the theoretical results and the setting of Habitat in the individual response.

 We thanks again for all the reviewers putting time and care into reviewing the paper, and we had answered all the reviewer's questions and minor comments in the individual response. To facilitate cross-referencing, we present all the references utilized in the response here

> Reference list
>
> [A] Savva, Manolis, et al. Habitat: A platform for embodied ai research. ICCV 2019
>
> [B] Chang, Angel, et al. Matterport3d: Learning from rgb-d data in indoor environments. arXiv 2017
>
> [C] Taiga, Adrien Ali et al. On Bonus-Based Exploration Methods in the Arcade Learning Environment. ICLR 2020
>
> [D] Wang Y et al. Efficient potential-based exploration in reinforcement learning using inverse dynamic bisimulation metric. NIPS 2024
>
> [E] Henaff, Mikael, et al. Exploration via elliptical episodic bonuses. NIPS 2022
>
> [F] Henaff, Mikael, et al. A study of global and episodic bonuses for exploration in contextual mdps. ICML 2023
>
> [G] Mikayel Samvelyan et al. MiniHack the Planet: A Sandbox for Open-Ended Reinforcement Learning Research. NIPS 2021
>
> [H] Ostrovski, Georg, et al. Count-based exploration with neural density models. ICML 2017
>
>  [I]    Zhang, T et al. Noveld: A simple yet effective exploration criterion. NIPS 2021

---

### Author Response · Authors · 2024-08-12

Dear reviewers,

Thank you once again for investing your valuable time in providing feedback on our paper. We had provided additional experiments and other responses based on your insightful suggestions. We highly appreciate the effort to provide in-depth reviews that helped us to improve our work.  Since the discussion period between the author and reviewers is rapidly approaching its end, we kindly request you to review our responses to ensure that all concerns are addressed and look forward to possibly receiving reply from you.

Best,

Authors

---

### Decision · Program_Chairs · 2024-09-25

**Decision:**

Accept (spotlight)

**Comment:**

All four reviewers agree that this is a solid paper, making significant contributions on an important topic, presented with clarity and extensive analysis.